# Deciphering local and regional hydroclimate resolves contradicting evidence on the Asian monsoon evolution

Annabel Wolf [1,2] ✉, Vasile Ersek [2] ✉, Tobias Braun [3], Amanda D. French[4], David McGee [5], Stefano M. Bernasconi [6], Vanessa Skiba [3], Michael L. Griffiths [7], Kathleen R. Johnson [1], Jens Fohlmeister [8], Sebastian F. M. Breitenbach [2], Francesco S. R. Pausata [9], Clay R. Tabor [10], Jack Longman [2,11], William H. G. Roberts[2], Deepak Chandan [12], W. Richard Peltier[12], Ulrich Salzmann[2], Deborah Limbert[13], Hong Quan Trinh[14] & Anh Duc Trinh[15]

The winter and summer monsoons in Southeast Asia are important but highly variable sources of rainfall. Current understanding of the winter monsoon is limited by conflicting proxy observations, resulting from the decoupling of regional atmospheric circulation patterns and local rainfall dynamics. These signals are difficult to decipher in paleoclimate reconstructions. Here, we present a winter monsoon speleothem record from Southeast Asia covering the Holocene and find that winter and summer rainfall changed synchronously, forced by changes in the Pacific and Indian Oceans. In contrast, regional atmospheric circulation shows an inverse relation between winter and summer controlled by seasonal insolation over the Northern Hemisphere. We show that disentangling the local and regional signal in paleoclimate reconstructions is crucial in understanding and projecting winter and summer monsoon variability in Southeast Asia.

The Asian winter monsoon is commonly associated with cold and dry northwesterly winds across East Asia and northeasterlies in Southeast Asia[1]. While not associated with significant rainfall over most regions, the winter monsoon delivers substantial rainfall in some coastal regions, such as Vietnam, the Philippines, Southeast India, Sri Lanka and Japan, where it plays a critical role in agriculture, water resources and natural hazard risks related to flooding and landslides. These regions include some of the world's largest food producers and exporters, causing not only the regional economy but also the already precarious global food trade to be vulnerable to changes in winter

[1]Department of Earth System Science, University of California, Irvine, CA 92697, USA. [2]Department of Geography and Environmental Sciences, Northumbria University Newcastle, Newcastle-upon-Tyne NE1 8ST, UK. [3]Potsdam Institute for Climate Impact Research, 14473 Potsdam, Germany. [4]Environmental Research Institute, Waikato University, Hamilton 3240, New Zealand. [5]Department of Earth, Atmospheric and Planetary Sciences, Massachusetts Institute of Technology, Cambridge, MA 02139-4307, USA. [6]Department of Earth Sciences, ETH Zürich, Zürich 8092, Switzerland. [7]Department of Environmental Science, William Paterson University, Wayne, NJ 07470, USA. [8]Federal Office for Radiations Protection, 10318 Berlin, Germany. [9]Department of Earth and Atmospheric Sciences, Centre ESCER (Étude et la Simulation du Climat à l'Échelle Régionale) and GEOTOP (Research Center on the dynamics of the Earth System), University of Quebec in Montreal, Montréal, QC, Canada. [10]Department of Earth Sciences, University of Connecticut, Storrs, CT 06269, USA. [11]Institute for Chemistry and Biology of the Marine Environment (ICBM), University of Oldenburg, Oldenburg 26129, Germany. [12]Department of Physics, University of Toronto, 60 St. George Street, Toronto, ON M5S1A7, Canada. [13]Oxalis Company, Phong Nha Bố Trạch, Quảng Bình, Viet Nam. [14]Institute of Chemistry, Vietnam Academy of Science and Technology, Ha Noi 10072, Viet Nam. [15]Nuclear Training Center, Vietnam Atomic Energy Institute, 140 Nguyen Tuan, Thanh Xuan, Ha Noi 11416, Viet Nam. ✉e-mail: wolfa2@uci.edu; vasile.ersek@northumbria.ac.uk

monsoon rainfall. In contrast to the very well-studied Southwest Summer Monsoon (SWM), there are no robust records documenting the long-term changes in Southeast Asian rainfall associated with the Northeast Winter Monsoon (NEM) under pre-industrial conditions. Therefore, the timing and mechanisms of hydroclimate variability in the region over longer timescales are not very well understood. Consequently, many climate models underestimate observed NEM rainfall by as much as 50%, leaving considerable uncertainty in future climate projections. These uncertainties are a harmful knowledge gap impacting planning for food production and geohazard management across this broad region[1].

Modern meteorological observations show a strong coupling between the Asian winter and summer monsoons[2]. However, while several paleoclimate studies have found a positive correlation over glacial-interglacial timescales[3–5], there is still no consensus on whether these two seasonal monsoon systems were similarly linked during the Holocene[6–15]. Several studies tried to address the co-evolution of the winter and summer monsoon in Southeast Asia over the Holocene, arguing for[16] and against[10,17,18] an inverse correlation, while others found that this relation changes depending on time scales[13,19]. Interestingly, solar insolation was argued to be the main driver of both a positive and negative correlation[10,16,17]. Thus, the nature and drivers of Holocene summer and winter monsoon variability and their co-evolution in Southeast Asia remain unclear and additional paleoclimate records are needed.

One of the issues often overlooked is the regional and local component of the Southeast Asian monsoon system. Both the winter and summer monsoon are characterized by large-scale circulation patterns common across the broad Asian monsoon region, which can be traced via oxygen isotopes in precipitation[20]. However, the timing of peak local precipitation is highly variable and oxygen isotopes fail to capture the local hydroclimate in many locations[20,21]. This is particularly true for coastal central Vietnam which does not receive substantial rainfall during the summer monsoon season like the rest of mainland Southeast Asia (MSEA; Fig. 1a, b). This is due to the Trường Sơn mountains, which act as an orographic barrier and create a rain shadow effect that limits the impact of southwesterly-sourced moisture during summer[20,22,23]. Instead the region receives over 75% of its annual rainfall between September to November from the western Pacific (Fig. 1c), and the effective rainfall (difference between rainfall and evapotranspiration) is positive from September to December only (Fig. 1d). Autumn and winter rainfall in Vietnam is related to the northeasterly winds of the NEM circulation sourced from the Pacific (Fig. 1b), unlike the SWM, which is sourced from the Bay of Bengal (Fig. 1a[20]). Therefore, paleoclimate records from central Vietnam allow us to reconstruct NEM variability and to test its relationship with the SWM.

Central Vietnam is likely one of the most extreme examples for the decoupling of the local and regional signal: lows and highs in oxygen isotope ratios occur during summer and winter respectively, but peak rainfall is in autumn. However, this decoupling of local (rainfall

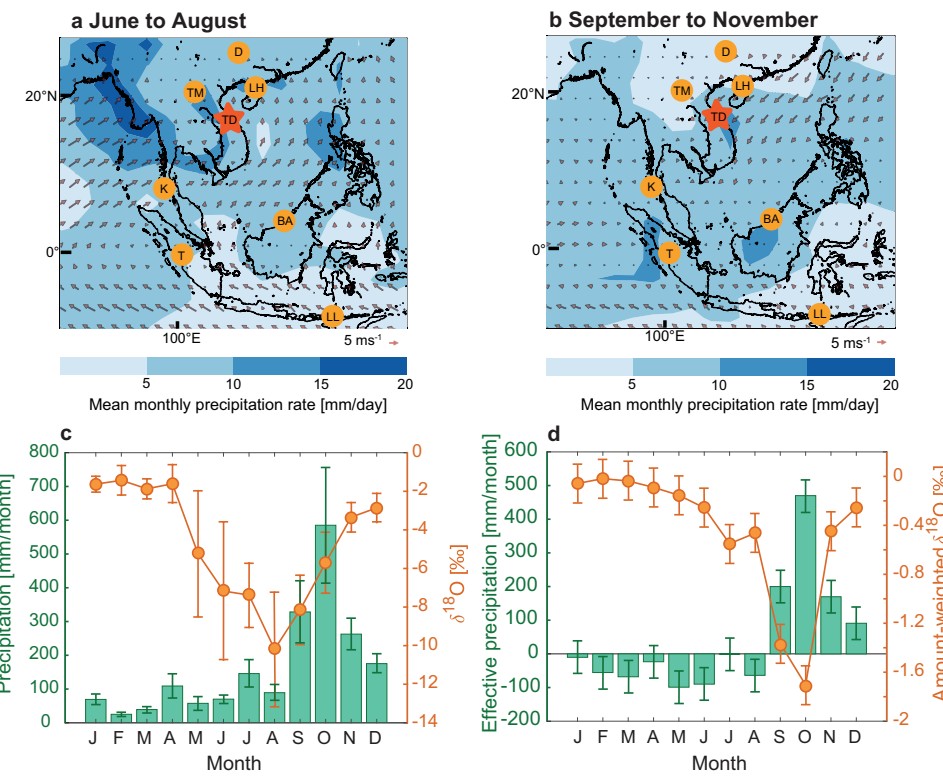

**Fig. 1 | General circulation and rainfall patterns in Southeast Asia.** Precipitation and circulation patterns during (**a**) the summer monsoon season and (**b**) the autumn/winter monsoon season (NEM) in Southeast Asia (red star indicates GNIP (Global Network of Isotopes in Precipitation) station in Dong Hoi central Vietnam, ca. 70 km from Thien Duong Cave used in this study). During summer, strong southwesterlies prevail over mainland Southeast Asia, leading to intense rainfall during the summer monsoon. Between mid-September and December strong northeasterly winds carry moisture into central Vietnam. $\delta^{18}$O (VSMOW) and rainfall recorded at the GNIP station in central Vietnam (**c**), showing low values in summer and high values in winter. Rainfall and $\delta^{18}$O data are from 2014 to 2021 and error bars give standard error of the mean. Effective precipitation and amount-weighted $\delta^{18}$O were calculated from the GNIP data and TerraClimate dataset[88] **d**. Precipitation is shown in green and $\delta^{18}$O in orange. Shading in (**a**) and (**b**) shows long-term mean monthly precipitation rate from the GPCP (Global Precipitation Climatology Project) Version 2.3 Combined Precipitation Dataset[89], covering 1981 to 2010. The arrows indicate surface wind strength at 10 m using the u and v components of the ERA5 data set[90]. Circles show the location of proxy records discussed in this work: **D** Dongge Cave (China)[91], **LH** Lake Huguang Maar (China)[16], **TM** Tham Doun Mai Cave (Laos)[43], **TD** Thien Duong Cave (Vietnam, red star), **K** Klang Cave (Thailand)[92], **BA** Bukit Assam Cave (Borneo)[41], **T** Tangga Cave (Sumatra)[25,74], **LL** Liang Luar Cave, Flores (Indonesia)[93].

amount) and regional (oxygen isotopes) components within the monsoon also occurs across other regions of Southeast Asia[20,21,24,25]. Deconvolving the correlation between the SWM and NEM can only be addressed if both components are considered and studied in detail across different time scales. We present a record of NEM variability during the Holocene using a speleothem from central Vietnam and distinguish between the regional circulation, reflected in oxygen isotopes, and local rainfall amount via carbon isotopes and trace elements. We apply a proxy system modeling approach to disentangle the regional component in our oxygen isotope record from the local signal. Using a Monte-Carlo principal component analysis, we investigate the correlation between NEM and SWM on a local and regional scale. Further, we compare the co-evolution of NEM and SWM during the mid-Holocene, a time with pronounced changes in local hydroclimate and use climate modeling to understand the mechanisms behind this shift in summer and winter monsoon.

## Results and discussion

### Multiproxy records of Southeast Asian monsoon precipitation

We use oxygen and carbon isotope ratios ($\delta^{18}O$ and $\delta^{13}C$) and high-resolution trace elements (Mg, Sr, Ba, P, U, Al, Si) measured in stalagmite TD3 (Fig. S1), recovered from Thien Duong Cave, central Vietnam (17.5°N, 106.5°E) in 2015 (Fig. 1) to reconstruct the NEM rainfall and circulation patterns in Southeast Asia over the Holocene. In central Vietnam, $\delta^{18}O$ in precipitation has a pronounced seasonal cycle, with low values during summer (June to August) and high values during the rest of the year, similar to other parts of MSEA[20,26,27]. This seasonal cycle in $\delta^{18}O$ is controlled by the shift between moisture sources, which include the Indian Ocean during summer (low $\delta^{18}O$) and the Pacific Ocean (high $\delta^{18}O$) during the rest of the year[20]. Seasonal rainfall patterns in central Vietnam differ strongly from the observed cycles in $\delta^{18}O$, with a peak in rainfall between September and December (Fig. 1c). Therefore, central Vietnam offers the opportunity to reconstruct regional circulation patterns and local rainfall forced by the NEM. While the former is reflected in the speleothem $\delta^{18}O$ signal, the latter is recorded by speleothem $\delta^{13}C$ and Mg/Ca.

Here, we interpret $\delta^{13}C$ and Mg/Ca variability as primarily driven by prior calcite precipitation (PCP), a process commonly associated with co-variance between $\delta^{13}C$ and Mg/Ca. When PCP occurs, $^{12}CO_2$ is preferentially degassed and magnesium, as well as other trace elements with partition coefficients smaller than 1, is preferentially excluded from calcite precipitated in the epikarst or on stalactite surfaces, leading to increased $\delta^{13}C$ and Mg/Ca in the remaining solution which ultimately feeds the stalagmite[28]. PCP is enhanced during dry periods, as has been shown by previous studies[29–32]. Therefore, we interpret increased Mg/Ca and $\delta^{13}C$ values in TD3 as indicative of relatively dry conditions at our cave site (Fig. S1a−e). This interpretation is further supported by Principal Component Analysis (PCA), which reveals that Mg/Ca, Sr/Ca and Ba/Ca co-vary in TD3, consistent with a common PCP control (Fig. S1f).

While several studies have interpreted speleothem trace elements and carbon isotopes as proxies for PCP and local water balance[33], recently, it has been recognized that PCP can also alter the oxygen isotopic composition of cave waters[28], and hence speleothem calcite. Similar to $\delta^{13}C$, $\delta^{18}O$ is subject to Rayleigh-based isotope enrichment during calcite precipitation[34] enhanced with increasing PCP, especially if PCP occurs at the cave ceiling[28]. Previously, it was believed that such an enrichment of the oxygen isotope ratio in drip water is sufficiently counteracted by $HCO_3^- \leftrightarrow H_2O$ buffering with the water reservoir[35]. However, more recent studies showed that this buffering is limited by lengthy exchange times between the bicarbonate and the water reservoir[36] and that oxygen isotopes might be significantly increased by PCP. In order to distinguish between PCP-induced changes in $\delta^{18}O$ and the initial rainfall $\delta^{18}O$ signal, we estimate the extent of PCP which occurred throughout the reconstructed time period based on the

Mg/Ca variability and a Rayleigh-based oxygen isotope evolution model[37–40]. We conduct the correction procedure with varying input parameters (temperature and $pCO_2$) to estimate uncertainties of our approach (Fig. 2a). The simulation using 426 ppmv and 21 °C (see Methods) is assumed to be the most accurate and is used as the main timeseries, whereas the remaining 12 time series are used as uncertainties. Details about cave monitoring and the $\delta^{18}O$ correction can be found in the Methods section. We apply a Rayleigh-model to estimate the length of PCP for each time point with corresponding Mg/Ca values and use it to correct the speleothem $\delta^{18}O$ record for the PCP-induced $\delta^{18}O$ changes. We then test the impacts the correction has on our analysis using a synthesis of $\delta^{18}O$ records across Southeast Asia. Because of the uncertainties inherent to our $\delta^{18}O$ correction due to simplifying cave internal processes, we focus on large-scale trends in the time series.

We interpret the corrected $\delta^{18}O$ record of TD3 as reflective of changes in the moisture source related to the NEM[20], and thus a measure of NEM strength. The corrected $\delta^{18}O$ record of TD3 shows lower $\delta^{18}O$ values during the mid-Holocene and higher values in the early and late Holocene (Figs. 2f and S6). This pattern agrees with previously reported speleothem records from Borneo, indicating decreasing $\delta^{18}O$ values in the mid-Holocene[41] (Fig. 2g). This trend has been interpreted as a result of enhanced convection over this region, broadly following autumn insolation in the Northern Hemisphere[42]. The $\delta^{18}O$ records from northern MSEA and Sumatra instead show decreased values during the early Holocene and an increasing trend after (Fig. 2d, e and h), which is consistent with summer rather than autumn insolation.

The lower $\delta^{18}O$ values observed in TD3 and the Borneo record suggest an intensification of the NEM during the mid-Holocene, in contrast to local hydroclimate proxies (Mg/Ca and $\delta^{13}C$) in TD3 and other paleoclimate records (Figs. S2; S6) which suggest pronounced drying during this time. The dry period started at 5 ka and ended abruptly at 3.2 ka, interrupted by two brief wet episodes at ~4.3 and ~3.7 ka. A dry period is also observed in other speleothem and lake records from Southeast Asia indicating widespread dry mid-Holocene conditions (Fig. S2). Specifically, Mg/Ca ratios and carbon isotopes in a speleothem record from Laos (Fig. S2a) show a pronounced local drought, beginning around 5 ka and lasting until around 3 ka[43]. Contemporaneously, a speleothem from southern Thailand exhibits a 2000-years-long hiatus, potentially indicating reduced water supply to sustain speleothem growth in the south of MSEA (Fig. S2c). While the $\delta^{18}O$ record from Borneo suggests strong convection over this region from 5 ka to 3 ka, as with our $\delta^{18}O$ record in central Vietnam (Figs. 2a and S2d), the $\delta^{13}C$ record from Borneo reveals a period of increased values at 4 ka, potentially indicating locally dry conditions. Two speleothem records from the Maritime Continent indicate a shift towards dry conditions around 5 ka in Sumatra (Fig. S2e) and Flores (Fig. S2f). Dry conditions during this time have also been reported from lake sediments in northern Thailand[44], Cambodia[45,46], and southern China[47]. Our data ($\delta^{13}C$ and Mg/Ca) shows that NEM precipitation declined during the mid-Holocene, while our $\delta^{18}O$ record suggests a more distal moisture source, potentially reflecting enhanced NEM winds. Here, we aim at resolving this apparent contradiction. Firstly, we collate a dataset of proxy records across Southeast Asia and study their coherence with clear implications on common forcing mechanisms. As opposed to coherent local SWM and NEM rainfall variability, their regional trends are anti-correlated. Secondly, we underpin these findings on the interplay of regional vs. local drivers based on model simulations.

### Holocene monsoon coherence across Southeast Asia

To investigate the spatiotemporal pattern of regional monsoon variability across Southeast Asia during the Holocene and to compare our proxy record in Vietnam with nearby records, we synthesized

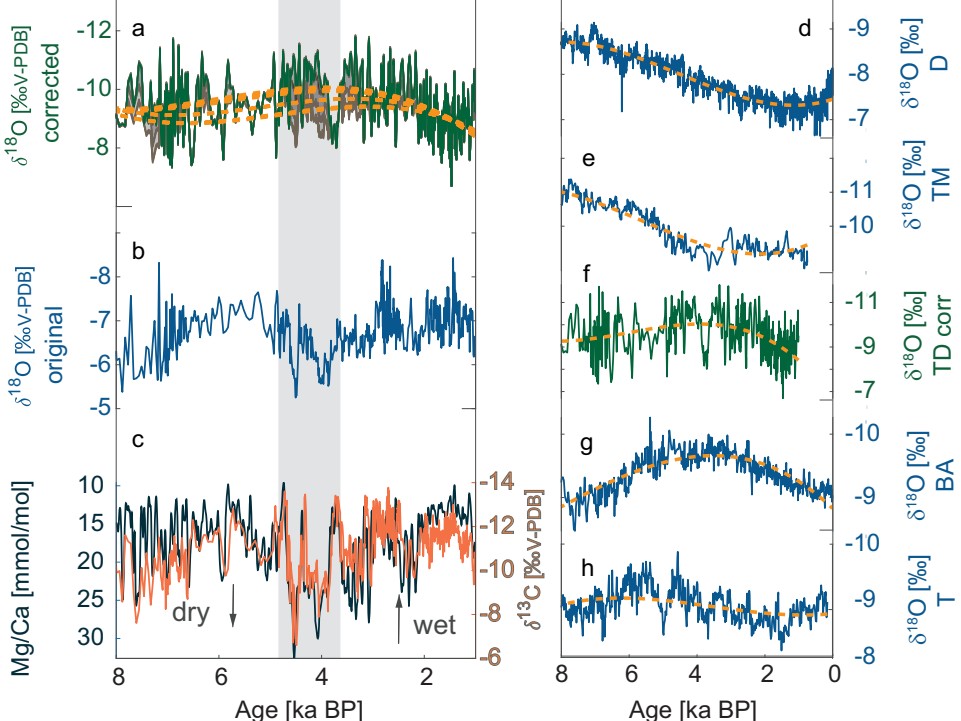

**Fig. 2 | Multi-proxy record obtained from stalagmite TD3 and regional $\delta^{18}O$ records. a** PCP (prior calcite precipitation) corrected (426 ppmv, 21 °C) and **b** original $\delta^{18}O$ values for stalagmite TD3, with (**a**) including uncertainties in gray and cubic fits (yellow lines) for each simulation. **c** $\delta^{13}C$ (orange) and Mg/Ca (black) published speleothem values for TD3, with a drought period between 5 to 3.5 ka indicated by grey vertical shading. **d–h** Oxygen isotope records from stalagmites across Southeast Asia (blue lines) and cubic fit (yellow lines). Cave names: **D** Dongge, **TM** Tham Doun Mai, **TD** Thien Duong, **BA** Bukit Assam, **T** Tangga.

published speleothem $\delta^{18}O$ records in Southeast Asia (Fig. S3) by using Monte-Carlo Principal Component Analysis (MC-PCA[48], see Methods). To evaluate the impact of our PCP-corrected $\delta^{18}O$ time series on the MC-PCA results and estimate the reliability of our approach we perform the MC-PCA on both the uncorrected (Fig. S4) and corrected (Fig. S5) $\delta^{18}O$ record of TD3. Other $\delta^{18}O$ records used in this analysis were not corrected for PCP, because the authors of the original studies found limited or no evidence for PCP, except for the Tham Doun Mai record in Laos[43]. This record shows no correlation between $\delta^{13}C$ and $\delta^{18}O$, suggesting that either there was sufficient time to reach equilibrium after PCP or other processes might have dominated the isotopic signatures in this cave.

The leading Principal Component (PC1) of $\delta^{18}O$ records explains 57% of the variance when TD3 is not corrected for PCP (Fig. S4a). PC1 exhibits a strong positive loading on the Laos (Tham Doun Mai Cave) (0.55) and China (Dongge Cave) (0.56) records, and a modest loading of 0.26 on the Sumatra (Tangga Cave) record. These results suggest a positive relation between these records which means they co-varied (increased or decreased synchronously) over the Holocene. The Tangga Cave record and other speleothem records from India and East Asia showed a high coherence over glacial-interglacial timescales[25], which appear less pronounced during the Holocene in our results. By contrast, PC1 displays a strong negative relation with Borneo (Bukit Assam Cave) (−0.48) and a modest negative relation with Vietnam (Thien Duong) (−0.29) (Fig. 3a). The orbital trend in the five $\delta^{18}O$ records is aligned with summer and autumn insolation (Fig. 3g): Here, we compare the long-term trends in the $\delta^{18}O$ records with temporal changes in insolation and find that records loading strongly positive on PC1 follow summer insolation most closely, while those with a negative correlation to PC1 follow autumn insolation. The long-term trends of each $\delta^{18}O$ time series used for the PCA shows significant strong correlations with seasonal insolation for each site (Fig. 3g), further evidencing this link. We conclude that seasonal insolation is a main driver of $\delta^{18}O$ across Southeast Asia for both the SWM and NEM. $\delta^{18}O$ in all records becomes more negative when insolation is increased, however, autumn and summer insolation peak at different times during the Holocene, potentially causing the apparent anti-correlation (Figs. 3 and 4). Modern observations and model simulations indicate that decreased $\delta^{18}O$ in rainfall over Southeast Asia is linked to enhanced convection, more distal moisture source, and increased rainout along the trajectory[20,21,49]. Thus, we propose that higher insolation leads to increased rainout and the moisture travel distance from the source, which results in decreasing $\delta^{18}O$ values in both the SWM and NEM.

Interestingly, the majority of our PCP-corrected time series (presented as the cubic fits, Fig. 3g) closely follows insolation, however, those corrections with high cave air $pCO_2$ (3000 and 2000 ppmv) are deviating the most. Values much higher than 1600 ppmv (highest modern cave air $pCO_2$ is ~1600 ppmv) are unlikely to have been reached over the Holocene, due to the cave being well-ventilated and changes in atmospheric $CO_2$ over the Holocene only varied between ~240 to 270 ppmv (by 30 ppmv)[50]. While unlikely, it should be noted that an increase in cave air $pCO_2$ to 2000 and 3000 ppmv alters the long-term trend of $\delta^{18}O$ and reduces the correlation between the cubic fit of $\delta^{18}O$ and SON insolation to $r = -0.69$ (for 3000 ppmv, 21 °C), compared to $r = -0.92$ (for 426 ppmv, 21 °C). Considering the uncertainties in our correction approach and recognizing that cave environment processes are simplified, we find that our results are a promising step in accounting for PCP influence on speleothem $\delta^{18}O$, which can enable the separate investigation of local and regional dynamics in our proxy record.

The overall positive loading on the PC2 of $\delta^{18}O$ indicates that all records share, to some extent, a common forcing. Indeed, PC2 exhibits a strongly-to-moderately positive loading on Sumatra (0.80), Vietnam (0.36), and Borneo (0.42), while weak relationship exist with Laos (0.06) and China (Dongge) (0.10) (Fig. 3b). Thus, after removing the trends contained in PC1, PC2 is more consistent with the local

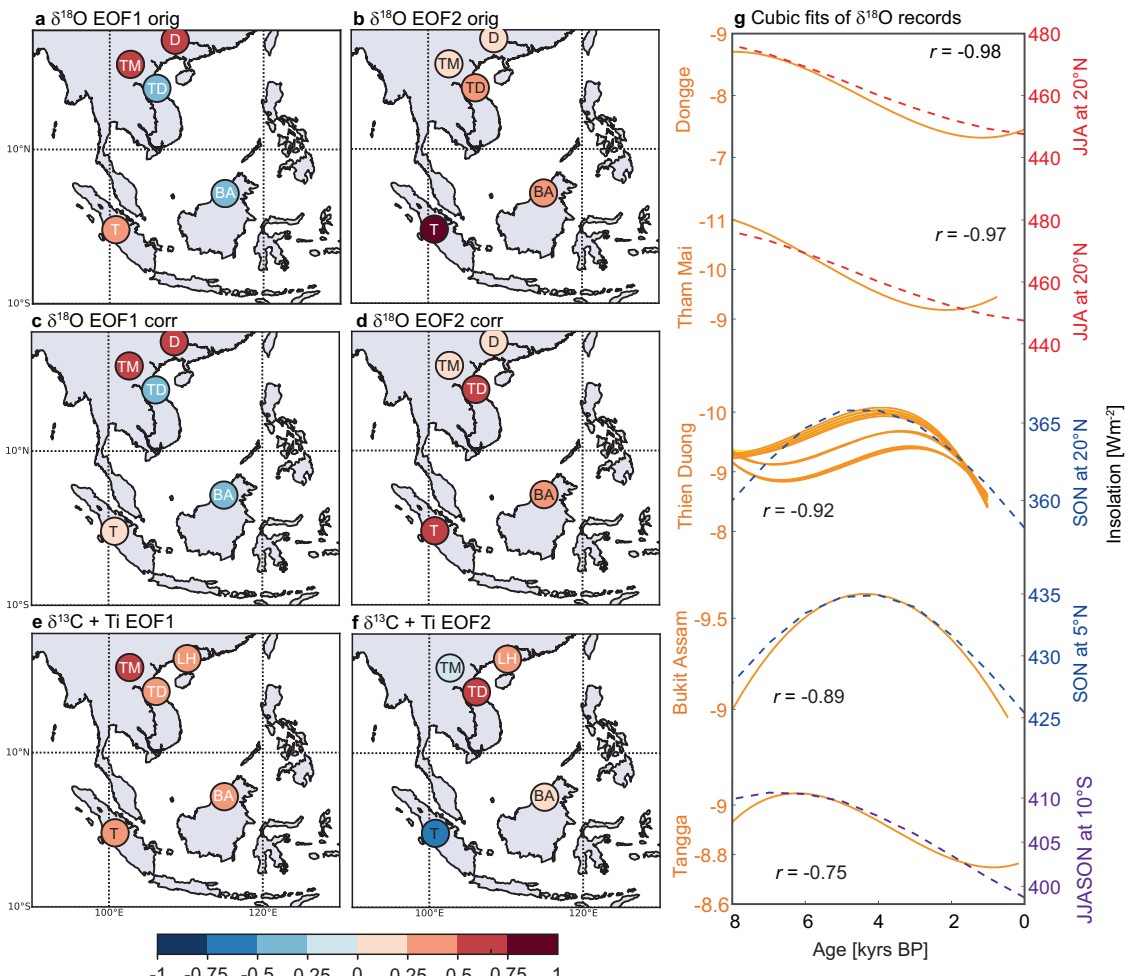

**Fig. 3 | EOF and PCs for speleothem records in Southeast Asia.** EOFs for original $\delta^{18}O$ on PC1 (**a**) and PC2 (**b**), EOFs for corrected $\delta^{18}O$ on PC1 (**c**) $\delta^{18}O$ of PC2 (**d**), and for local proxies ($\delta^{13}C$) for PC1 (**e**) and PC2 (**f**). EOFs on PC2 of $\delta^{18}O$ and PC1 of $\delta^{13}C$ show that the millennial-scale spatial pattern is in-phase across Southeast Asia, whereas the orbital pattern (**a**) and (**c**) indicate spatial heterogeneity. Panels (**a**) to (**d**) include only the $\delta^{18}O$ record from Dongge Cave because the $\delta^{13}C$ record is not available. In (**e**) and (**f**) the titanium record of lake Huguang Maar (**LH**) is used as substitute instead. **g** cubic fits for the records used in panel (**c**), including the PCP (prior calcite precipitation)-corrected time series of TD3. *r*-values are given for the correlation between cubic fits and insolation (dashed lines) at each location, all correlations are significant with *p*-values below 0.05. For TD3, *r*-values are given for the main PCP-corrected timeseries with $p$CO$_2$ of 426 ppmv and 21 °C. Cave names: **D** Dongge, **TM** Tham Doun Mai, **TD** Thien Duong, **BA** Bukit Assam, **T** Tangga.

hydroclimatic pattern (Fig. 4c), however, also contains some of the autumn insolation trends reflecting the Vietnam and Borneo time-series (Fig. 4b). When comparing the timeseries of PC2 of $\delta^{18}O$ and PC1 of $\delta^{13}C$, we also find that PC2 of the uncorrected $\delta^{18}O$ time series shares more similarities with PC1 of the carbon records (Fig. 4c). This high-lights that our correction indeed has the potential to remove the local hydroclimatic (PCP) signal. The mismatch between local and regional signals of the Southeast Asian monsoon is likely one of the limiting factor in deconvolving summer and winter monsoon variability across this region in past studies.

In order to investigate the local components of the Southeast Asian monsoon, we now apply the MC-PCA on $\delta^{13}C$ speleothem records and one lake record to compensate for the $\delta^{13}C$ from Dongge cave which is not publicly available. Results of the PCA for speleothem $\delta^{13}C$ records and the titanium content of Lake Huguang Maar[14] from southern China reveal that all records are strongly to moderately positive correlated with PC1 (Fig. 3e). The strongest loading between PC1 and the proxy records is found for Laos (0.62), followed by Sumatra (0.46), and Borneo (0.41) (Fig. 3e and S7). The winter mon-soon records—e.g. the stalagmite from Vietnam (TD3) and the Lake Huguang Maar record from southern China—show a weaker, but also

positive relation with PC1 (Vietnam: 0.27 and southern China: 0.37). This suggests that the first PC derived from stalagmite $\delta^{13}C$ records and a lake record from Southeast Asia likely share a common forcing throughout the Holocene, despite the different modern rainfall regimes unique to each site (Fig. 1c, d[20]). However, the records more dominated by winter monsoon circulation also show a stronger load-ing on PC2, with 0.57 for Vietnam and 0.43 for southern China (Huguang Maar). Additionally, Sumatra shows a strong negative rela-tion with PC2 of −0.59, which can be explained by local hydroclimate, characterized by minimum rainfall in boreal summer and strong influence of the Indo-Australian summer monsoon[25]. The Borneo record loads most heavily on PC3 (−0.81). This may be due to the local hydroclimate with year-round rainfall or, alternatively, to the dom-inance of non-hydrologic controls on speleothem $\delta^{13}C$ in Borneo (Bukit Assam Cave)[51]. Nevertheless, the first three PCs combined explain 77% of the variance, thus covering most of the variability present in all five records. Crucially, the dry conditions observed in the proxy records between -5 to -3 ka are evident in both PC1 and PC2, suggesting strong coherence between the sites during this time (Fig. S7). Furthermore, these results highlight that the dry period observed in TD3 and other paleoclimate records, affected most of Southeast Asia, with reduced

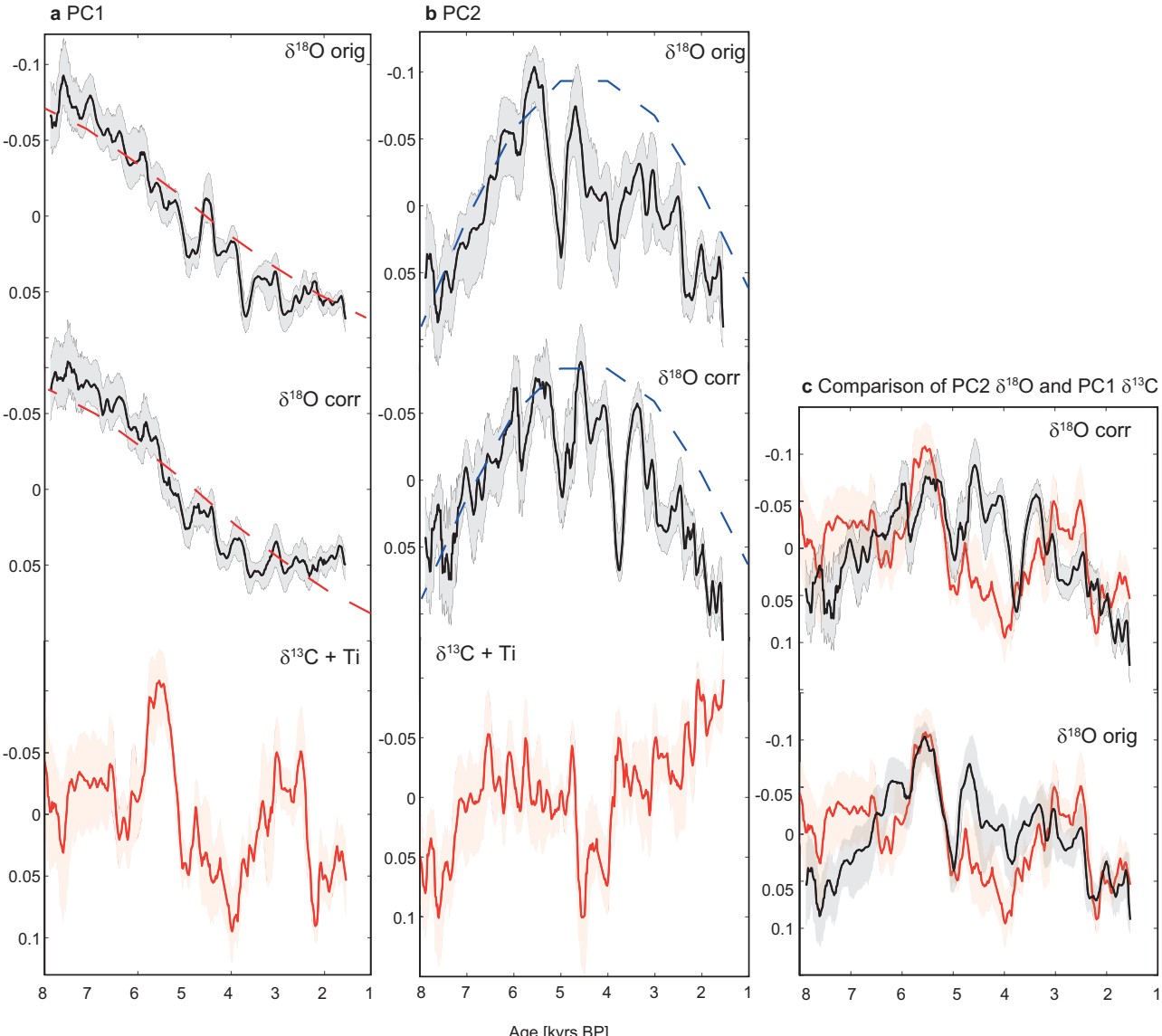

**Fig. 4 | PCs for speleothem and lake records in Southeast Asia.** PC1 (**a**) and PC2 (**b**) timeseries of the original, corrected $\delta^{18}$O and $\delta^{13}$C + Ti records. Black lines show oxygen and orange lines show carbon isotope ratios, with age uncertainties in shading. The uncorrected PC2 ($\delta^{18}$O) shares similarities with PC1 of the $\delta^{13}$C + Ti records (**c**). Dashed lines show summer (red) and autumn (blue) insolation.

rainfall confined not just to the summer monsoon season, but also to the autumn/winter season. In the next step we investigate the forcing mechanisms of this dry period and their effects on the regional and local aspects of the Southeast Asian monsoon.

**Monsoon coherence during the mid-Holocene**

The mid-Holocene is characterized by large-scale changes in tropical climate conditions and mean ocean states–e.g. reduced ENSO variance, pronounced droughts or monsoon intensification (Figs. S8 and S9a). Based on our results, Southeast Asian speleothem records suggest substantial local drying during this period, whereas the regional NEM intensity (reflected in speleothem $\delta^{18}$O) seems to have increased (Figs. 2 and S2). We investigate this apparent mismatch by using three state-of-the-art climate models and synthesize results of mid-Holocene idealized sensitivity experiments from EC-Earth[52], iCESM[53], and the University of Toronto version of CCSM4 (UofT CCSM4;[54]) climate models. These model experiments simulate mid-Holocene climate following PMIP protocol, but additionally account for changes in vegetation cover across the Saharan region, a forcing which has

recently been suggested to affect local SWM rainfall[43]. For all modeling experiments, we compare results from two sensitivity experiments for the mid-Holocene (6 ka). (1) MH$_{ORB}$ simulations with mid-Holocene insolation, greenhouse gases, vegetation cover and dust load based on the PMIP3 (EC-Earth) or PMIP4 (iCESM and UofT CCSM4) protocols. (2) MH$_{GS}$ simulations have a prescribed vegetated 'Green Sahara' (GS), additionally, EC-Earth and iCESM have a reduced dust load. Further details on the varying methods and parametrizations incorporated in these different modeling experiments can be found in Pausata et al. (2017; EC-Earth)[52], Tabor et al., (2020; iCESM)[53], and Chandon and Peltier et al., (2020; UofT CCSM4)[54]. In summary, we compare runs simulating the vegetation over the Sahara as pre-industrial with simulations using a Green Sahara, to isolate the effect of dust loading (MH$_{ORB}$-MH$_{GS}$) during the mid-Holocene.

Starting with the regional Southeast Asian monsoon, we use iCESM simulations for the mid-Holocene, to examine the impact of changing mid-Holocene climate on Asian rainfall $\delta^{18}$O variability (Figs. 5 and 6). During summer, rainfall throughout southern China and MSEA exhibits significantly higher $\delta^{18}$O values (Fig. 5a). Simulated

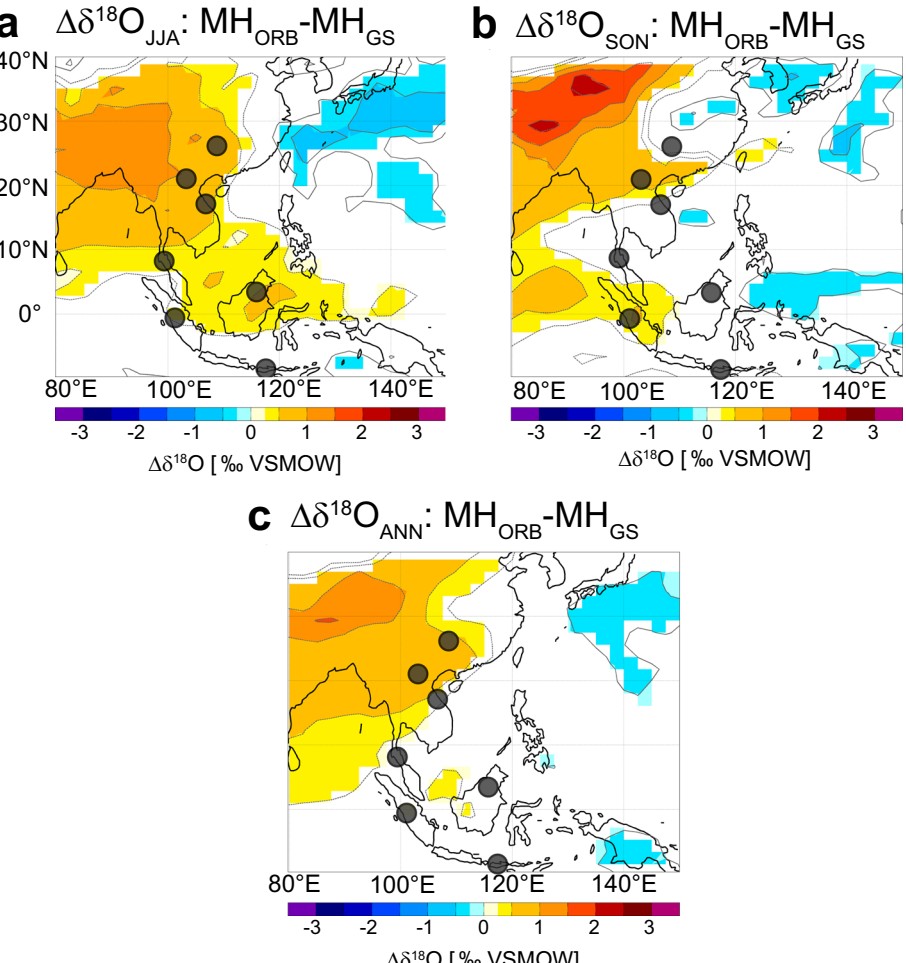

**Fig. 5 | Simulated $\delta^{18}O$ changes during the mid-Holocene.** Changes in $\delta^{18}O$ for (**a**) summer, (**b**) autumn and (**c**) annual mean between the mid Holocene without (MH$_{ORB}$) and with (MH$_{GS}$) Saharan vegetation as simulated by iCESM. Thus, MH$_{ORB}$-MH$_{GS}$ represents reduced Saharan vegetation and increased dust at the end of the Green Sahara. Shadings highlight only changes significantly different at the 5% level using a local (gridpoint) $t$ test. The contours follow the color bar intervals (solid for positive and dashed for negative anomalies; the zero line is omitted). Dots indicate locations of proxy records described in Fig. 1.

autumn (SON) $\delta^{18}O$ values in rainfall across southern China/northern MSEA and Sumatra also increase, which is in agreement with proxy observations over the Holocene (Figs. 3a, c and 5b). While absolute values cannot be compared between the model and the proxy records, the spatial patterns concur: during the mid-Holocene the $\delta^{18}O$ proxy records can be grouped by trends: low values peak during the mid-Holocene in Borneo and Vietnam (Figs. 3g and 5b), whereas records from southern China/northern MSEA and Sumatra show continuously decreasing trends from the early to the late Holocene (Figs. 3g and 5a). The same spatial pattern can be seen in iCESM (SON) $\delta^{18}O$ values, indicating an increase in $\delta^{18}O$ values in northern MSEA and Sumatra only. These results suggest that the inverse relation between SWM and NEM evident over the entire Holocene (Fig. 3) was not only present but likely enhanced during the mid-Holocene due to vegetation changes in the Sahara.

In terms of rainfall, all three models show widespread drying in Southeast Asia during the Mid-Holocene, with EC-Earth showing the largest changes (Fig. 6). The EC-Earth model indicates decreased summer and autumn rainfall across much of Southeast Asia in a scenario of reduced Saharan vegetation and higher dust load relative to the Green Sahara state, contributing to the overall decline in annual rainfall (Fig. 6). While EC-Earth agrees with the dry conditions inferred from paleoclimate proxies across Southeast Asia, UofT CCSM4 suggests wetter summer conditions during the mid-Holocene

in northern Southeast Asia, contrasting proxy observations from Laos and southern China (Fig. S2). Here, we compared the changes in the local proxy record before (8–5 ka BP) and after (5–3.5 ka BP) the end of the Green Sahara, where all proxy records used in the MC-PCA suggest drier conditions during 5 to 3.5 ka BP (Fig. 4a, b). During autumn the UofT CCSM4 model simulates reduced precipitation in northern MSEA and Sumatra and wetter conditions in parts of Borneo and no significant change in annual mean precipitation over MSEA. Combined, the mean of all three models suggests drying during autumn and a lower annual mean in the north of MSEA and southern Southeast Asia (Fig. 6 bottom panel).

While there is some disagreement among the models, the EC-Earth simulations generally agree with our interpretation of reduced rainfall in MSEA. Specifically, reduced Saharan vegetation and increased dust forcing result in an 'El Niño-like' mean state across the Pacific (Fig. S9b). Such an El Niño-like mean state is associated, with weakened easterly trade winds leading to warmer central/eastern Pacific SSTs, and increased easterlies in the eastern Indian Ocean resulting in lower SSTs off Sumatra (Fig. S9b). Upwelling in the South China Sea and the East China Sea could potentially inhibit convective activity over the moisture source region during the peak of the northeast winter monsoon season in Vietnam (Fig. S9b), contributing to the much lower average rainfall observed in our proxy records. Our simulations and proxy synthesis show that rainfall during the mid-

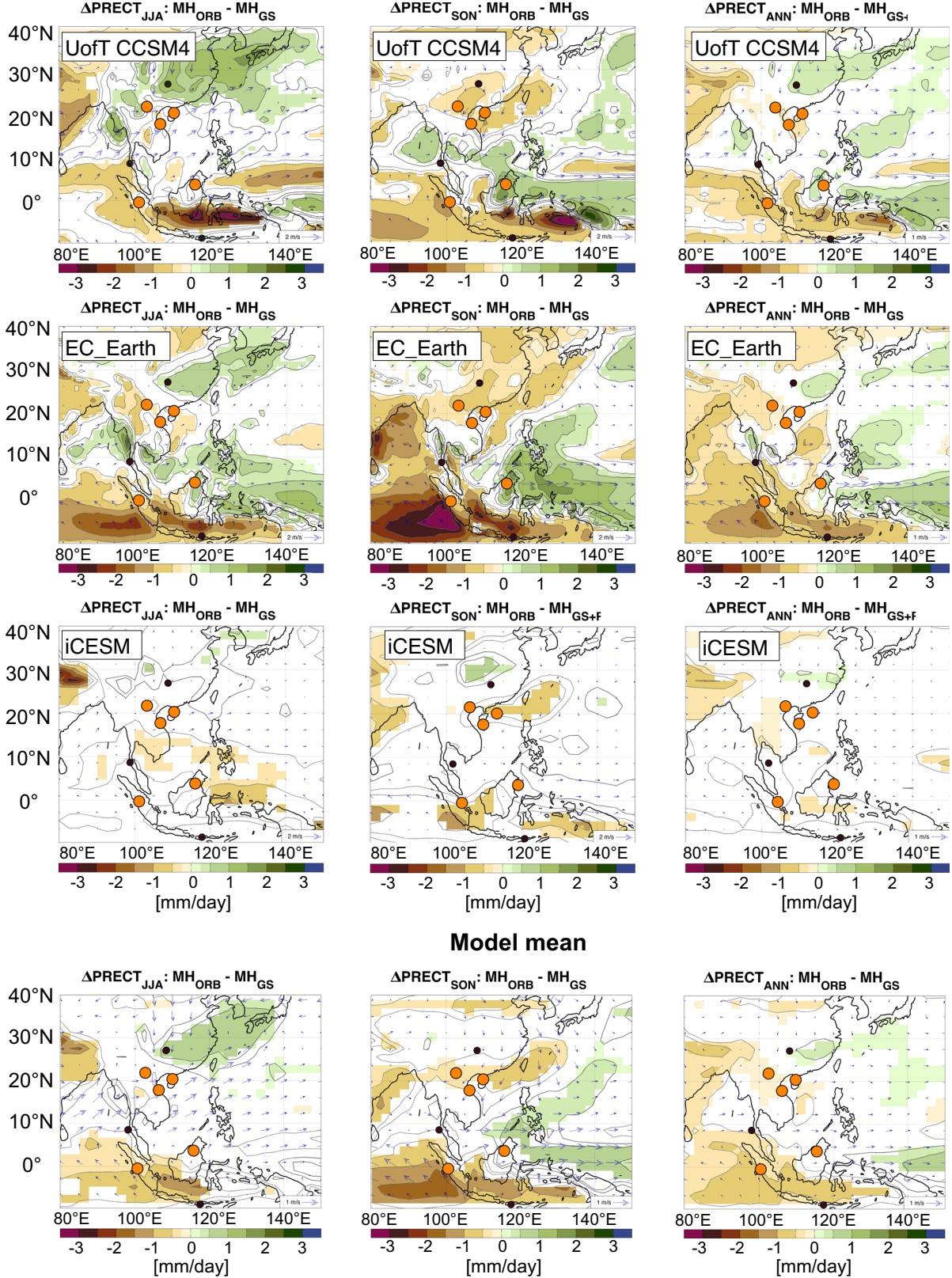

**Fig. 6 | Mid-Holocene simulation of local precipitation.** Simulated precipitation changes between MH$_{ORB}$ (Mid-Holocene orbital) and MH$_{GS}$ (Mid-Holocene Green Sahara) of the University of Toronto version of NCAR CCSM4 model (UofT CCSM4; top panels), EC Earth (middle panel) and the iCESM (bottom panel) considering the effect of the end of the Green Sahara and associated increase in dust load. Model mean is shown in the panels below. Dots show locations of proxy records, with small dots not included in the MC-PCA (Monte-Carlo Principle Component Analysis) and larger dots included. The larger yellow dots indicate drier conditions in the interval 3.5–5 ka compared to the interval 5-8 ka BP (Figs. S2 and S3).

Holocene was reduced in summer, autumn and year-around (annual mean), not simply during the summer monsoon as previously suggested[43].

Interestingly, the mid-Holocene spatial $\delta^{18}O$ pattern simulated in iCESM agrees well with spatial $\delta^{18}O$ patterns of proxy records in the PCA. However, iCESM results indicate no significant changes in precipitation amount, despite pronounced hydrological changes observed in the proxy records during the mid-Holocene. Thus, $\delta^{18}O$ and precipitation share no direct link in iCESM, a result which is also found in the deviation between $\delta^{18}O$ and $\delta^{13}C$ records from MSEA. These are significant results because climate models still have limits to accurately simulate local rainfall in the tropics, due to the inability to properly account for convection. The simulation of $\delta^{18}O$ appears to be more reliable in our results than rainfall. This highlights that local rainfall and the regional monsoon signal preserved in $\delta^{18}O$ need to be considered in both climate model simulations and proxy reconstructions when investigating summer and winter monsoon relations in Asia.

Here, we addressed the long-term co-evolution between summer and winter monsoon in Asia over the Holocene by reconstructing NEM rainfall variability. We employed speleothem-based proxies from central Vietnam to infer local hydroclimatic and regional monsoon changes over the last ca. 8 ka. We showed that SWM and NEM rainfall in Southeast Asia strongly covaried over most of the Holocene, which is in good agreement with climate model simulations (Fig. 6). As the main driver for monsoon coherence between sites in Southeast Asia, we identify eastward shifts of the Walker Circulation as regulators of rainfall over the region (Fig. S9b). The tight positive coupling of the local rainfall from SWM and NEM stands in contrast to the regional components, which shows an inverse relation between SWM and NEM driven by changes in insolation in the Northern Hemisphere over the Holocene. Thus, their main forcing mechanisms are internal, via ocean-atmosphere feedbacks and external via insolation, respectively. Our results highlight the need for careful distinction between the local and regional aspects of the Southeast Asian monsoon, and inform on the choice of proxies when aiming to reconstructs monsoon variability.

## Methods
### Uranium-thorium dating
Hand drilled powder samples from TD3 weighing 250–300 mg were prepared for U-Th dating at the Massachusetts Institute of Technology following Edwards et al. (1987)[55]. Samples were dissolved in nitric acid and spiked with a $^{229}Th – ^{233}U – ^{236}U$ tracer, followed by isolation of U and Th by iron co-precipitation and elution in columns with AG1-X8 resin. The isolated U and Th fractions were analyzed using a Nu Plasma II-ES multi-collector inductively coupled plasma mass spectrometer (MC-ICP-MS) equipped with an Aridus 2 desolvating nebulizer. Reported errors for $^{238}U$ and $^{232}Th$ concentrations are estimated to be ±1% due to uncertainties in spike concentration; analytical uncertainties are smaller than 1%. Ages were corrected for detrital $^{230}Th$ assuming an initial $^{230}Th/^{232}Th$ atomic ratio of $(4.4 \pm 2.2) \times 10^{-6}$ and the age was reported in years before present, where the present is defined as 1950 C.E. Decay constants for $^{230}Th$ and $^{234}U$ were taken from Cheng et al. (2013)[56] and, the decay constant for $^{238}U$ of $1.55125 \times 10^{-10}$ yr$^{-1}$ [57] is used. The depth-age model was calculated in COPRA[58] based on 21 U-Th dates, from which 6 have been excluded by stratigraphic order method.

### Trace elements
Trace element measurements were made using a RESOlution-SE Compact 193 nm excimer laser ablation (LA) system in tandem with an Agilent 8900 Inductively Coupled Plasma Mass Spectrometer (ICP-MS) at Waikato University, New Zealand. Analyses were conducted by pulsing the laser at 15 Hz with a 50 µm spot size and energy density of 5 J/cm² at a scan speed of 26.118 µm/s. Measurements were made inside

the milled track to ensure that stable isotope and trace elemental data have the same depth-age relation. Ultra-high purity helium is used as the carrier gas to deliver the ablated sample from the LA system to the ICP-MS. The ICP-MS was optimised to maximum sensitivity daily using the NIST 612 glass standard. Background counts (gas background, measured with the laser off) were collected for 40 seconds between samples, allowing for complete washout. NIST glass standards (610, 612) were analysed after each sample track to account for any instrumental drift. Data processing was performed using Iolite (v3.32;[59]). Background counts are subtracted from the raw data and all data were standardised to NIST 612 glass certified reference values. NIST 610 glass was utilized as a secondary standard. GeoReM database[60] is utilized for NIST glass certified values.

### Stable isotopes
In total 371 low-resolution (1 mm) and 853 high-resolution samples (200 µm) were drilled and analyzed. Carbonate stable isotope analyses were performed using a Delta V Advantage isotope ratio mass spectrometer (IRMS) (Thermo Fisher Scientific) interfaced with a Gas Bench II and a CTC GC PAL autosampler at ETH in Zürich (high-resolution) and Northumbria University (low-resolution). Results were normalized against the international NBS18 and NBS19 standards and in-house standard Wiley ($\delta^{18}O = -7.2$, $\delta^{13}C = -0.41$), and reported in permil (‰) relative to VPDB. Carbonate was dissolved using phosphoric acid and the sample weight was between 70 to 100 µg.

### Mineral determination
Fourier-transform infrared spectroscopy (FTIR) was used to determine carbonate mineralogy at Northumbria University, Newcastle. 3 mg of stalagmite powder was analyzed on a PerkinElmer Spectrum RX I FTIR over a spectrum of 640-4200 cm$^{-1}$ with 32 replications. The detection limit was about 5 wt% and the measurement had a precision of approximately ±10% resolution is 0.8 cm$^{-1}$. Before each run the background spectrum was measured and automatically subtracted from the sample spectrum. TD3 shows a typical calcite FTIR spectrum[61], with pronounced peaks at 712 cm$^{-1}$ and 872 cm$^{-1}$.

### PCP removal
Similarity of Mg/Ca and $\delta^{13}C$ together with MC-PCA analysis reveals that Mg/Ca of TD3 is a proxy for PCP. PCP-influenced bicarbonate of drip water is not representing the initial rainfall $\delta^{18}O$ signal because it is modified by Rayleigh-induced isotopic enrichment. Mg/Ca can provide information on the extent of PCP and its variation over time and, thus, can be used to remove the PCP signal from the speleothem $\delta^{18}O$. With this approach it is feasible to obtain a record of initial bicarbonate $\delta^{18}O$ of cave drip water, which in turns allows determination of the $\delta^{18}O$ of infiltration/rain water.

To estimate PCP length from the Mg/Ca time series it is necessary to calculate calcite precipitation from a solution, which we approximate by using an exponential approach[62]. In addition, we use a solely Rayleigh-enabled isotope evolution model[37-40,63] to simulate $\delta^{18}O$ evolution while $CO_2$ degasses and $CaCO_3$ precipitates. In the present study, we extend this approach by implementing the behaviour of the Mg/Ca ratio with evolving $CaCO_3$ precipitation. Skiba & Fohlmeister, 2023[40] found that changes in drip interval result in only small deviations in the isotope composition of $CaCO_3$ in comparison to the potential effect of variations in the duration of PCP. As the evolution of the Mg/Ca ratio of a solution is treated similar to the isotope evolution in the model (both are described by a Rayleigh approach), the influence of drip interval changes on the Mg/Ca ratio of speleothem $CaCO_3$ can also be considered small compared to those induced by changes in the duration of PCP. Thus, we neglect changes in isotopic composition which are caused by variations in drip interval here.

To simulate calcite precipitation, the evolution of Mg/Ca and the isotope composition over time, we use a modern cave temperature of

21 °C[64] which is assumed to have remained relatively constant throughout the last 8000 yrs of speleothem growth. This is a reasonable assumption as cave temperatures mainly reflect mean annual air temperatures and the Holocene is known to have had a relatively constant temperature history in the (sub-) tropics. Furthermore, the model needs additional parameters such as an initial $Ca^{2+}$ concentration of drip water, $Ca^{2+}_{initial}$. This parameter depends mostly on soil or karst $pCO_2$ and the equilibrium $Ca^{2+}$ concentration, $Ca^{2+}_{eq}$, with respect to the cave air $pCO_2$. During monitoring of Thien Duong cave in spring (2015, 2016) and summer (2016), cave air $CO_2$ values of ~420 ppmv to ~1580 ppmv have been measured[64]. This corresponds to a range of $Ca^{2+}_{eq}$ values of ~0.7 to ~1.1 mol/m³. We use the minimum value of 0.7 mol/m³, which corresponds to a $pCO_2$ of 426 ppmv. This value represents the lower endmember of observed cave air $pCO_2$, where carbonate precipitation, i.e. speleothem growth, is enhanced compared to periods of elevated cave air $CO_2$ levels. When using a $Ca^{2+}_{eq}$ of 0.7 mol/m³, we need to set the $Ca^{2+}_{initial}$ to a minimum value of ~2.73 mol/m³ to explain the range of Mg/Ca observed in the speleothem data. This value corresponds to an equilibrium $pCO_2$ value of ~25,300 ppmv, which is one order of magnitude larger than observed in present-day soil air $pCO_2$ (i.e. 3400 and 1600 ppmv in spring 2015 and 2016, respectively;[64]). However, there is growing evidence that the air in karst systems can have much higher $pCO_2$ values than soils[32,65–68]. Therefore, we consider the $Ca^{2+}_{initial}$ value of 2.73 mol/m³ or more as realistic and simulate two cases: 1) $Ca^{2+}_{initial}$ of 2.73 mol/m³, i.e. in equilibrium with a $pCO_2$ of ~25,300 ppmv and 2) $Ca^{2+}_{initial}$ of 3.18 mol/m³, i.e. in equilibrium with a $pCO_2$ of ~40,000 ppmv. Similar to temperature, we assume that modern $CO_2$ dynamics did not change substantially over the last 8000 yrs and keep the $Ca^{2+}_{initial}$ and $Ca^{2+}_{eq}$ parameters constant throughout the simulation.

During calcite precipitation, the Mg/Ca ratio of the solution changes according to the partition coefficient, $D_{(Mg)}$, between the solution and precipitated calcite (e.g., Johnson et al., 2006[69]). We use a $D_{(Mg)}$ of 0.015, calculated for T = 21 °C[33] which is based on laboratory experiments performed under cave analogous conditions.

To account for the possible variations in cave air temperature and $pCO_2$, we simulate 12 additional PCP corrections using modern 21° ± 2, which is the Holocene range of cave air temperatures reconstructed from a cave in Borneo[70] and $pCO_2$ ranging from 260 to 3000 ppmv.

As the spatial resolution of the Mg/Ca record is higher than the $\delta^{18}O$ record, we need to downsample the Mg/Ca record to the coarser timescale of the $\delta^{18}O$ in order to use both proxies in our calculations. For downsampling, we use Gaussian kernels (Gaussian kernel width = 0.5 mm, which is the maximum integrated depth during sampling of carbonate powder for isotope measurements) and calculate smoothed Mg/Ca values for each time step for which $\delta^{18}O$ is available. In absence of values for the initial Mg/Ca ratio of the solution, we identify the smallest Mg/Ca value from the smoothed record and assume that this value represents the initial Mg/Ca value. In other words, we assume that the Mg/Ca value at $t_0 = 0$ s, when no PCP has occurred (Mg/$Ca_0$ = 8.3 mmol/mol) was constant through time and forms the starting point for the coupled calcite precipitation—element ratio—isotope composition simulation. After the drip water comes in contact with cave air, calcite precipitates with a decreasing trend while Mg/Ca and $\delta^{18}O$ of the solution are increasing (Fig. S10).

From the speleothem Mg/Ca values, we can estimate the duration of PCP necessary to obtain each Mg/Ca value based on estimated Mg/$Ca_0$ to calculate the PCP-induced $\delta^{18}O$ change. For example, if the duration for PCP is ~650 s ($t_1$ in Fig. S10) and $Ca^{2+}_{initial}$ is 2.73 mol/m³, an increase in Mg/Ca of the solution by ~15 mmol/mol (from Mg/$Ca_0$ = 8.3 mmol/mol to the Mg/Ca value of the speleothem sample of 23.3 mmol/m³) is required. This corresponds to an increase in $\delta^{18}O$ of ~4.1‰, which is applied as a correction to the $\delta^{18}O$ speleothem record. The two model set-ups 1) $Ca^{2+}_{initial}$ of 2.73 mol/mol³ and 2) $Ca^{2+}_{initial}$ of 3.18 mol/mol³ lead to similar results.

## Monte-Carlo Principle Component Analysis (MC-PCA)

To calculate the spatiotemporal coherence between 'n' records across Southeast Asia, an age uncertainty-aware Principal Component Analysis (PCA) is applied. The approach largely follows Deininger et al. (2017)[48], using Monte Carlo sampling of age model realizations to adjust for the age uncertainties in all records[71]. Firstly, all speleothem age models have been recalculated in COPRA using the initial published U-Th ages. With COPRA, 2000 distinct time axes compatible with age uncertainty are sampled employing a Monte-Carlo sampling scheme and subsequently transferred into uncertainty of the proxy values[58]. This results in a single 'error-free' time axis and 2000 distinct proxy realizations which are compatible with the U-Th dating uncertainties. PCA requires constant sampling intervals in order to compute and decompose the covariance matrix of normalized proxy time series. Thus, all records are resampled with a constant sampling interval of 75 years and an overlap of 80% between consecutive windows to ensure an equal sampling resolution, adapting to the lowest resolved record. As proposed by Deininger et al. (2017)[48], Gaussian Kernel-based smoothing is applied in this step to align the temporal variability between the differently resolved proxy records. Smoothing windows are centered at the resulting regularly sampled dates. Smoothed proxy realizations are z-normalized, yielding zero mean and unit standard deviation. Finally, PCA is applied to each of the 2000 sets of smoothed and z-normalized proxy realizations. Since every proxy realization can be regarded as a shifted and stretched/compressed version of the underlying 'real' proxy time series, each set of proxy realizations yields an individual characterization of spatiotemporal coherence represented by m ≤ n modes. Each mode consists of a principal component (PC) and n EOFs (empirical orthogonal functions, also called 'loadings'). EOFs are normalized such that they are equivalent to the linear correlation between the respective PC and a proxy realization. As discussed by Deininger et al. (2017)[48], a smoothing and a flipping effect can be observed in the MC-PCA. In order to obtain meaningful estimates of the median and confidence limits for the PCs and EOFs, the flipping effect requires additional processing of the ensemble of results. Firstly, an intrinsic sign ambiguity exists for the singular value decomposition used to numerically solve the eigen decomposition of the data covariance matrix[72]. Small deviations in one of the 'n' proxy records can suffice to result in a 'random' sign flip of the resulting PCs and EOFs, irrespective of the MC-based procedure. This will in some cases result in a bimodal distribution of the EOFs for a given PC. Secondly, a sign flip can result from age uncertainty whenever a set of proxy realizations from the age model ensembles is analyzed that is comprised of anti-phased instead of phased signals as discussed in Deininger et al. (2017)[48]. In this work, k-means clustering is applied for each ensemble of PCs to distinguish flipped and non-flipped solutions[73]. In this context, another effect was observed when age uncertainty of any of the included proxy records is large: apart from the flipped and non-flipped solution for the 1st PC, a third distinct solution can be identified which exhibits strong similarity to the 2nd PC. This results from an overlap between the variance explained by each of the modes and may occur between any given pair of ith and i+1th PC if dating uncertainties are large. While in most cases (e.g. 1800 out of 2000) the 1st mode is represented by strongly correlated (flipped and non-flipped) PCs, in fewer cases variance may be maximized by what is largely identified as the 2nd mode. Such a scenario is identified successfully by the proposed k-means clustering of PCs with k = 3 and should be considered in future application of the MC-PCA method. In this work, dating uncertainties were found to be sufficiently low such that this effect does not result in a significant number of 'swapped' modes. Consequently, EOFs and PCs of the flipped modes identified by k-means clustering are sign-flipped and a single median PC/EOF is computed from the respective ensemble of solutions for each mode. Age uncertainty is propagated by computation of the 2σ-error bars of the respective PC and explained variance. The statsmodel package was

used to compute PCA solutions for each set of proxy realizations and the sklearn package was used for k-means clustering. The MC-PCA is applied to $\delta^{13}C$ and $\delta^{18}O$ records respectively form Tham Mai Cave (Laos)[43], Thien Duong Cave (Vietnam) this study, Tangga Cave (Sumatra)[74], Bukit Assam Cave (Borneo)[41]. Additionally, a lake record from Lake Huguang Maar in China[14] was included. However, COPRA is specifically built for speleothem records, therefore the original age model without age uncertainties in all 2000 realization was used. For the $\delta^{18}O$ MC-PCA, Dongge Cave record was included instead of the Lake Huguang Maar lake record, but no $\delta^{13}C$ record was accessible. For both applications, three modes were computed motivated by their interpretability.

### Climate model simulations
For this work, three climate simulations were adapted to understand the forcing mechanisms of the mid-Holocene. For more details on the methodology used, refer to Pausata et al. (2017)[52] and Gaetani et al. (2017)[75] for EC-Earth, to Chandan and Peltier (2020)[54] for the UofT CCSM4 experiments and to Tabor et al. (2020)[53] and Thompson et al. (2021)[76] for the iCESM simulations.

### EC-Earth
Numerical simulations of the mid-Holocene were run using the EC-Earth version 3, with the atmospheric model based on the Integrated Forecast System (IFS cycle 36r4) (European Centre for Medium-range Weather Forecasts), including the H-TESSEL land model. The simulations were run at T159 horizontal spectral resolution, corresponding to ca. 1.125° by 1.125° and at a vertical resolution of 62 vertical levels. As an ocean model version 3.3.1 of the Ocean General Circulation Model-NEMO was used[77]. The model solves the primitive equations discretized on a curvilinear horizontal mesh with a horizontal resolution of ca. 1° × 1° and 46 vertical levels. The model is coupled every model hour with the Louvain-la-Neuve Ice Model-LIM370 at the surface, having the same horizontal resolution as NEMO.

Two experiments were conducted with both set to an orbital forcing corresponding to values at 6 ka, which were computed internally following Berger (1978)[78]. Greenhouse gases accord with the PMIP3/CMIP5 protocol and other boundary conditions were set pre-industrial values following to the PMIP/CMIP5 protocol. The first experiment, the MH$_{ORB}$ experiment, uses the dust climatology of the MERRAero data as the Community Atmosphere Model, based on the long-term monthly mean (1980–2015)[79], simulating a desert over the Sahara region. In the second experiment (MH$_{GS}$), the vegetation type over the Sahara domain (11° to 33°N and 15°W to 35°E) is set to shrub and the dust amount is reduced by up to 80%[52] simulating a vegetated Sahara, as proposed by paleoclimate data[80]. Changing vegetation cover from shrub (MH$_{GS}$) to desert (MH$_{ORB}$) corresponds to an increase in the surface albedo from 0.15 to 0.30 and a decrease in the leaf area index from 2.6 to 0.2. The dust changes from the MH$_{GS}$ to MH$_{ORB}$ experiment correspond to an increase in the global total aerosol optical depth of 0.02[81]. To avoid abrupt transitions in dust concentrations, a smoothing filter was used and the dust reduction was applied evenly above 150 hPa, because, at these heights, aerosol particles are uniformly distributed. No accurate vegetation reconstruction for the mid-Holocene is available at present, thus the model cannot provide a realistic reconstruction, but give insights on potential feedback mechanisms during this time.

### University of Toronto CCSM4 (UofT CCSM4)
In addition to the EC-Earth, the University of Toronto version of the NCAR CCSM4 model (UofT CCSM4) is used to perform similar experiments[82]. Peltier and Vettoretti (2014)[83] modified the ocean component of the model, namely the Parallel Ocean Program version 2, to be more appropriate for paleoclimate simulations. It was first applied by these authors to the study of glacial climates and

subsequently used for studying the mid-Pliocene warm period in Chandan and Peltier (2017, 2018)[84,85] and the mid-Holocene in Chandan and Peltier (2020)[54]. A detailed description of the modifications to the ocean component can be found in Peltier and Vettoretti (2014)[83] and in Chandan and Peltier (2020)[54]. Similar to the EC-Earth experiments, a mid-Holocene simulation was performed using orbital parameters at 6 ka and concentrations of trace gases following PMIP4/CMIP6 protocol[86]. The UofT CCSM4 MH$_{GS}$ simulations follow the vegetation cover recommended by Otto-Bliesner et al. (2017)[86] and Pausata et al. (2016)[81], similar to the EC-Earth MH$_{GS}$ run. However, the Guineo-Congolian tropical forest belt was additionally extended northward. The change in surface albedo resulting from MH$_{GS}$ to MH$_{ORB}$ is 0.18 to 0.30, similar to the EC-Earth run.

### iCESM
We use a water isotopologue tracking-enabled version of the Community Earth System Model (iCESM1;[87]), with a fully coupled configuration of iCESM1 with 1.9 × 2.5° atmosphere (CAM5) and land (CLM4), and nominal 1° ocean (POP2) and sea ice (CICE4). Similar to the other model runs, two experiments are performed:

A mid-Holocene configuration with orbital forcing corresponding to values at 6 ka and greenhouse gases (264.4 ppm $CO_2$, 597 ppb $CH_4$, 262 ppb $N_2O$) as in PMIP4 (MH$_{ORB}$; Otto-Bliesner et al., 2017), and a mid-Holocene configuration with prescribed Saharan vegetation (MH$_{GS}$) (Fig. S11). In MH$_{ORB}$, vegetation coverage types come from the preindustrial default in CLM4 with agricultural land reverted to natural plant functional types. In MH$_{GS}$, PI zonal average land conditions in Africa at ~11°N is extended, including vegetation/soil types and leaf/stem area indices, to all land in the Sahara and Arabian Peninsula. The addition of vegetation in the mid-Holocene reduces mean annual aerosol optical depth from 0.50 to 0.05 and surface albedo from 0.30 to 0.15 over the region of North Africa with modified vegetation. Further information about the simulation can be found in Tabor (2020)[53].

### Data availability
All proxy data generated in this study are provided at https://zenodo.org/record/8271126.

### Code availability
The code used to conduct the PCA and the PCP removal, including instructions and example data, are accessible via Zenodo https://zenodo.org/record/8271126.

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

## Acknowledgements

We are grateful to Trương Văn Luyến and Cao Văn Tâm for their great support during fieldwork. We want to thank Nguyễn Phi Hùng for supporting us with cave monitoring, especially monthly water sample collection and Madalina Jaggi (ETH) for analytical support. Thanks to the management board of the Phong Nha–Ke Bang National Park for their support and guidance. Vietnam Foundation for Science and Technology Development RCUK-NAFOSTED (grant numbers NE/P014577/1) to A.D.T., NSF grants AGS-1804747 to C.R.T., NSF grant AGS-2103129 to K.R.J. and AGS-2103051 to M.L.G., Northumbria Ph.D. studentship to A.W. V.S. and J.F. acknowledge funding by Deutsche Forschungsgemeinschaft (grant FO 809/6-1). The EC-Earth model simulations were enabled by resources provided by the Swedish National Infrastructure for Computing (SNIC) at the National Supercomputer Centre (NSC) and Cray XC30 HPC systems at ECMWF. The research of WRP at the University of Toronto is supported by NSERC Discovery Grant A9627.

## Author contributions

Sample export: A.D.T. Sampling: A.D.T., H.Q.T., A.W., V.E. U-Th dating: D.M. LA-ICP-MS: A.F., J.L. PCA: T.B. IRMS: S.M.B., S.F.M.B., A.W., V.E. ISOmodl ($\delta^{18}$O correction model): V.S., J.F. Climate models: F.S.R.P., C.R.T., W.R.P., D.C. Visualization: A.W., T.B., F.S.R.P. Cave Monitoring: A.D.T., D.L. Supervision: V.E., W.H.G.R., M.L.G., S.F.M.B., K.R.J., U.S. Writing—original draft: A.W., T.B., V.S., M.L.G., S.F.M.B., V.E., W.H.G.R. Writing—review & editing: A.W., T.B., V.S., M.L.G., S.F.M.B., V.E., W.H.G.R., K.R.J., U.S., T.B., F.S.R.P., C.R.T., W.R.P., D.C., A.D.T., H.Q.T., A.F., J.L., S.M.B., J.F., D.M.

## Competing interests

The authors declare no competing interests.
