## [Peer Review File · Nature Communications]

Deciphering local and regional hydroclimate resolves contradicting evidence on the Asian monsoon evolutionREVIEWER COMMENTS

Reviewer #2 (Remarks to the Author):

Dr. Wolf et al. present a new continuous Holocene speleothem record (oxygen isotopes, carbon isotopes, and trace elements) from central Vietnam. They argue that this record can disentangle the large-scale circulation reflected by oxygen isotopes and local rainfall changes reflected by carbon isotopes and trace elements. They further use a Monte-Carlo principal component analysis to investigate the spatiotemporal changes in $\delta^{18}O_p$ and $\delta^{13}C$ records across Southeast Asia. Besides, they use mid-Holocene climate simulation results to explore the potential mechanism of the mid-Holocene anomalies in $\delta^{18}O_p$ and $\delta^{13}C$ observed from proxy records. They concluded that the synchronous rainfall change in Southeast Asia is forced by changes in the Pacific and the Indian Ocean, whereas the inverse winter and summer regional atmospheric circulation is controlled by seasonal insolation over the Northern Hemisphere.

While I find the new Holocene speleothem record to be a useful indicator of the Northeast Winter Monsoon in Southeast Asia and potentially provides an opportunity to disentangle large-scale circulation and local rainfall, the discussion about the spatiotemporal change of monsoon and precipitation in Southeast Asia and the underlying mechanism of NEM change during the Holocene are not well supported and sometimes unconvincing. Below, I highlight my major concerns and suggestions.

1. The features of $\delta^{18}O$, $\delta^{13}C$, and trace elements time series described in the paper are not very clear to me. For example, I think the “lower $\delta^{18}O$ values during the mid-Holocene and higher values in the early and late Holocene” (Line 158-159) are not evident, given the combination of changes in high-frequency variability and millennial-scale changes in mean value. I would suggest the authors use statistical methods to identify the timing of step shifts in $\delta^{18}O$, $\delta^{13}C$, and Mg/Ca. Some potential options include the environmental time series change point detection (EnvCpt) method (Beaulieu and Killick, 2018, Journal of Climate), and regime shift detection (Rodionov, 2004, GRL). These methods can detect the change point in a time series based on changes in its mean value and variance.

Besides, I think the cubic fitted curves of the $\delta^{18}O$ record from TM (Fig. 2E) and TD (Fig. 2F) could be problematic as they extend outside the range of the original data (e.g., 0-1ka),

The mismatch between $\delta^{18}O$ and $\delta^{13}C$ and trace elements in the central Vietnam speleothem record is the main motivation of this work, and the different long-term patterns in $\delta^{18}O$ records across the Southeast Asian revealed by the cubic fit is key evidence for the inverse relation between SWM and NEM. So, I think it is important to use appropriate statistical methods to detect shifts and the timing of shifts in the $\delta^{18}O$, $\delta^{13}C$, and trace elements time series.

2. The mechanisms of the orbital-scale reverse response in SWM and NEM are unclear to me. The authors found positive loading on the PC1 of $\delta^{18}O$ in southern China and Sumatra and negative loading on the PC1 of $\delta^{18}O$ in Borneo and Vietnam. Based on the significant correlation between boreal

summer and autumn insolation, they conclude that “Holocene insolation changes are the primary drivers of the long-term trend of $\delta^{18}\text{O}$ speleothem records across Southeast Asia” (Line 239-242). However, the significant correlation between PC1 and autumn insolation is insufficient to prove NEM recorded in Vietnam and Borneo mainly follows autumn insolation because the temporal evolution of the PC1 of $\delta^{18}\text{O}$ (long-term increasing trend from 8ka to 1ka) doesn’t capture the main features in the $\delta^{18}\text{O}$ records from Vietnam and Borneo (more depleted $\delta^{18}\text{O}$ during the mid-Holocene). In addition, the authors didn't clearly explain how summer and autumn insolation would influence the SWM and NEM, respectively.

3. It’s very interesting to see SWM and NEM show reverse response (reflected by the PC1 of $\delta^{18}\text{O}$) on orbital timescale, whereas coherent rainfall pattern across Southeast Asia was found on millennial timescales (reflected by the PC1 of $\delta^{13}\text{C}$). It’s worth to further investigating the underlying mechanisms of the different regional circulation ($\delta^{18}\text{O}$) and local rainfall ($\delta^{13}\text{C}$) on different timescales. But I don’t think the mid-Holocene time-slice model simulation results alone are sufficient to explain the difference in the response of $\delta^{18}\text{O}$ and rainfall on different timescales.

4. It’s not clear to me why the authors use “MHORB-MHGS” to discuss the mechanisms of $\delta^{18}\text{O}$ and precipitation changes during the mid-Holocene observed from proxy records.

The MHORB simulations incorporate mid-Holocene insolation, greenhouse gases, vegetation cover, and dust load, and MHGS simulations have a prescribed vegetated ‘Green Sahara’ (GS), additionally” (Line 287-290). It’s not clear to me what climate forcings would “MHORB-MHGS” indicate and why the authors use the difference between the ORB and GS runs to identify the mechanisms driving mid-Holocene anomalies. Another issue related to this is the argument about reduced Saharan vegetation could contribute to the mid-Holocene lower average rainfall observed in proxy records (Line 331-338, Fig. S9B). It’s not clear to me why “MHORB-MHGS” can specifically reveal the influence of Saharan vegetation. I would suggest the authors use the MHORB minus preindustrial (or early Holocene) to investigate the contribution of orbital forcing on mid-Holocene $\delta^{18}\text{O}$ and precipitation anomalies, and (MHGS - MHORB) minus preindustrial (or early Holocene) to evaluate the influence of Green Sahara.

Other comments:

Fig. 1: (1) Please change “mean daily precipitation per month [mm/day]” in Fig. 1A and 1B to “mean monthly precipitation rate [mm/day]”. (2) Please change TH to TD (Line 108).

Fig. 2: Please use appropriate statistical methods to detect shifts in $\delta^{18}\text{O}$, $\delta^{13}\text{C}$, and Mg/Ca. Please make sure the cubic fit in Fig. 2E and 2F can reveal the long-term trend in $\delta^{18}\text{O}$ from TM and TD, respectively.

Line 162-166: Please show changes in autumn and summer insolation in Fig. 2 to back up this statement.

Fig. 3: (1) Please add the time series of PC2 of $\delta^{18}\text{O}$ orig, $\delta^{18}\text{O}$ corr, and $\delta^{13}\text{C}$. Please reverse the color bar for EOF, so that the warm color indicates positive loading, and the cold color indicates negative

loading. (2) In Fig. 3A-3F, please change “PC1” and “PC2” to “EOF1” and “EOF2”.

Reviewer #3 (Remarks to the Author):

The authors present a new isotope and trace element record from speleothems from Vietnam. They claim that it is the first reconstruction of winter monsoon for the Holocene period, which is important for understanding the continent's paleoclimate. The data has high quality, and the comparisons with climate models were also addressed to reinforce their interpretation. The paper is well written, and this novel has the potential to be published in Nature Communications. However, I see a major problem with their correction on the d18O record. This correction compromises the quality of the data and the interpretation on the manuscript. See my concerns about this correction below. Apart from this, I have only minor comments to improve the manuscript.

1) The correction is based on the approach presented by Deininger et al. (2021), which shows that PCP can also affect the d18O in stalagmites. This is something that so far has been ignored in the literature. However, the implication of the PCP on d18O presented in Deininger et al. is mainly theoretical and still needs further validation before being applied to speleothem data records. Furthermore, no calibration studies with speleothem were done before.

2) For this correction, the authors assume temperature and CO₂ in the soil to be constant during the Holocene period, which is unrealistic. Temperature changes were small but not constant. Slight elevations of temperature associated with drier periods can significantly increase evaporation with all associated effects on the isotopes. But the big problem is with regard the CO₂. The monitoring of soil CO₂ over two years already shows variability between 430 ppmv and 1580 ppmv. This variability was probably more significant during the Holocene. Furthermore, long periods of drier or wetter conditions can affect the vegetation and soil above the cave, changing the CO₂ concentration in the soil, which is the primary source of the cave's CO₂. The proxies presented in this paper present high variability, indicating significant rainfall variability above the cave, which will also affect the CO₂ concentration in the soil.

3) The authors use d13C and Mg as a “pure” proxy for PCP, which is unrealistic. I agree that both are proxies for PCP, but PCP is not the only force on these proxies. In using Mg to correct d18O, the authors are not only removing the PCP signal but also adding noise to the d18O signal.

4) In this correction, the author assumes that all correlation between d18O and Mg and d13C is due to PCP. However, other factors provide a similar correlation between these proxies other than PCP. Therefore, in removing this dependency from the d18O, the author also removes additional information. For example, during a strong monsoon, the “amount effect” provides more negative values to the d18O, but the high rainfall during this event also decreases the PCP leading to lower values of Mg and d13C. Therefore amount effect and rainfall variability also drive d18O, d13C, and Mg in the same direction.

Therefore, using the Mg/Ca and correcting the d18O, the authors are not just removing the PCP effect on the d18O but also the amount effect.

5) The authors argue that they are removing the effect of PCP on the d18O. But the corrected record has much higher variability than the original record, making me believe that this "correction" adds more variability and noise to the records than filter PCP.

Minor comments:

Fig.1: How the amount-weighted d18O was calculated? Were monthly averages used?

Fig2: I suggest including the name of the caves in the figure or at least on the captions. It isn't enjoyable to go to another figure to see the name of the records presented in this figure.

Fig. 2: it is difficult to distinguish the scales at the y-axes between D and E.

The authors also measured the Sr/Ca and Ba/Ca with laser ablation. Why were these data not shown in the paper?

Line 258: The authors say "This suggests that the first PC derived from stalagmite $\delta^{13}\text{C}$ records and a lake record from Southeast Asia likely share a common forcing throughout the Holocene...". According to the authors, d13C in the stalagmites is connected to the local hydrology due to the PCP process. However, PCP is absent in lakes. d13C in lakes is related to the isotopic composition of the organic matter/vegetation. If d13C from stalagmite and lakes correlates, this is suggestive that the d13C from speleothems is also affected by the d13C from the vegetation (something ignored in this manuscript) and not that the d13C from the lakes is affected by PCP, which is impossible. How do the authors explain it?

Best regards,
Valdir F. Novello

REVIEWER COMMENTS

Reviewer #2 (Remarks to the Author):

Dr. Wolf et al. present a new continuous Holocene speleothem record (oxygen isotopes, carbon isotopes, and trace elements) from central Vietnam. They argue that this record can disentangle the large-scale circulation reflected by oxygen isotopes and local rainfall changes reflected by carbon isotopes and trace elements. They further use a Monte-Carlo principal component analysis to investigate the spatiotemporal changes in $\delta^{18}\text{O}_p$ and $\delta^{13}\text{C}$ records across Southeast Asia. Besides, they use mid-Holocene climate simulation results to explore the potential mechanism of the mid-Holocene anomalies in $\delta^{18}\text{O}_p$ and $\delta^{13}\text{C}$ observed from proxy records. They concluded that the synchronous rainfall change in Southeast Asia is forced by changes in the Pacific and the Indian Ocean, whereas the inverse winter and summer regional atmospheric circulation is controlled by seasonal insolation over the Northern Hemisphere.

While I find the new Holocene speleothem record to be a useful indicator of the Northeast Winter Monsoon in Southeast Asia and potentially provides an opportunity to disentangle large-scale circulation and local rainfall, the discussion about the spatiotemporal change of monsoon and precipitation in Southeast Asia and the underlying mechanism of NEM change during the Holocene are not well supported and sometimes unconvincing. Below, I highlight my major concerns and suggestions.

1. The features of $\delta^{18}\text{O}$, $\delta^{13}\text{C}$, and trace elements time series described in the paper are not very clear to me. For example, I think the “lower $\delta^{18}\text{O}$ values during the mid-Holocene and higher values in the early and late Holocene” (Line 158-159) are not evident, given the combination of changes in high-frequency variability and millennial-scale changes in mean value. I would suggest the authors use statistical methods to identify the timing of step shifts in $\delta^{18}\text{O}$, $\delta^{13}\text{C}$, and Mg/Ca. Some potential options include the environmental time series change point detection (EnvCpt) method (Beaulieu and Killick, 2018, Journal of Climate), and regime shift detection (Rodionov, 2004, GRL). These methods can detect the change point in a time series based on changes in its mean value and variance.

Besides, I think the cubic fitted curves of the $\delta^{18}\text{O}$ record from TM (Fig. 2E) and TD (Fig. 2F) could be problematic as they extend outside the range of the original data (e.g., 0-1ka). The mismatch between $\delta^{18}\text{O}$ and $\delta^{13}\text{C}$ and trace elements in the central Vietnam speleothem record is the main motivation of this work, and the different long-term patterns in $\delta^{18}\text{O}$ records across the Southeast Asian revealed by the cubic fit is key evidence for the inverse relation between SWM and NEM. So, I think it is important to use appropriate statistical methods to detect shifts and the timing of shifts in the $\delta^{18}\text{O}$, $\delta^{13}\text{C}$, and trace elements time series.

We thank the reviewer for the suggestion; we have updated the cubic fits using the polyfit and polyval functions in Matlab to ensure a more robust fitting to the time series boundaries. Additionally, we performed several change point and trend analyses which have been added as a figure to the supplementary material (Fig. 2 and 3 shown below; Fig. S6) using EnvCpt and BEAST. Both approaches support our interpretation and show that indeed TD3 $\delta^{18}\text{O}_{corr}$ values were lower during the mid-Holocene and higher during the early and late Holocene, as shown in panel 3 of Fig. 1 and 3 in this document. They also show that there are significant shifts towards increased values in the $\delta^{13}\text{C}$ and Mg/Ca during the onset and end of the drought.

Figure 1: EnvCPT of analysis our proxies records: Left panel $\delta^{13}\text{C}$, middle panel $\delta^{18}\text{O}$ original and right panel $\delta^{18}\text{O}$ corrected. The analysis identifies a significant shift in $\delta^{13}\text{C}$ and $\delta^{18}\text{O}$ original at the start and end of the mid-Holocene drought (highlighted by the grey shading). The right panel shows that after PCP correction, there is a significant shift in the time series towards lower values during the mid-Holocene (grey shading), which is interrupted by short positive excursions. These results strengthen our interpretation of lower $\delta^{18}\text{O}$ values during the mid-Holocene in our corrected $\delta^{18}\text{O}$ time series. Cpt = changepoint detection, AR=auto regression.

Figure 2: BEAST change point and decomposition for our local proxy record, the mid-Holocene drought is indicated in grey. Top panel (Y) shows the data points plotted as circles and the mean as line plot. The trend is also plotted in the second panel from the top for better visibility. Third panel show the probability of changepoint occurrences over time (Pr(tcp)). "tOrder" shows the time-varying polynomial order estimated to fit trend (close to zero, meaning a flat /constant line). "slpSign" is the probabilities of trend slope being positive (red), zero (green), or negative (blue).

Similar to the EnvCPT analysis the BEAST analysis also shows significant changes in our time series at the timing of the mid-Holocene drought. This further strengthens our interpretation of the time series to record this drought similarly to the reconstruction from Laos (Tham Duon Mai cave).

Figure 3: BEAST analysis for the corrected $\delta^{18}\text{O}$ time series, same caption as above.

The results identify changes towards lower values in the corrected $\delta^{18}\text{O}$ time series between 3.5 and 5 ka, interrupted by a short (~200 yearlong) increase, supporting our initial conclusion of lower values during this period, as also shown in the EnvCPT plots. Here, we would like to highlight that we are primarily interpreting the long-term trend only for the Holocene, which is revealed by the cubic fits in the main manuscript Fig. 2, 3 and 4.

2. The mechanisms of the orbital-scale reverse response in SWM and NEM are unclear to me. The authors found positive loading on the PC1 of $\delta^{18}\text{O}$ in southern China and Sumatra and negative loading on the PC1 of $\delta^{18}\text{O}$ in Borneo and Vietnam. Based on the significant correlation between boreal summer and autumn insolation, they conclude that “Holocene insolation changes are the primary drivers of the long-term trend of $\delta^{18}\text{O}$ speleothem records across Southeast Asia” (Line 239-242). However, the significant correlation between PC1 and autumn insolation is insufficient to prove NEM recorded in Vietnam and Borneo mainly follows autumn insolation because the temporal evolution of the PC1 of $\delta^{18}\text{O}$ (long-term increasing trend from 8ka to 1ka) doesn’t capture the main features in the $\delta^{18}\text{O}$ records from Vietnam and Borneo (more depleted $\delta^{18}\text{O}$ during the mid-Holocene). In addition, the authors didn’t clearly explain how summer and autumn insolation would influence the SWM and NEM, respectively.

To address Reviewer#2 concerns and improve clarity, we have added a new panel to Fig. 3 (G), which shows the newly performed cubic fits of $\delta^{18}\text{O}$ records used in the PCA, and compare them with insolation closest to the record’s location, considering the main season for peak rainfall. The summer monsoon records (Dongge and Tham Duon Mai) show similar trends to JJA insolation, while Thien Duong and Bukit Assam records share most similarities with SON insolation. The record from Tangga cave matches best with mean JJA and SON insolation, potentially due to the influence of the Australian monsoon. Here, we propose that an increase in insolation is strengthening the rainfall and associated winds, leading to longer rainout and more depleted $\delta^{18}\text{O}$ values reaching the cave sites. The anti-correlation of the $\delta^{18}\text{O}$ in rainfall associated with summer and winter monsoon thus results from the anti-correlation between insolation at these locations. We have added the following text to the main manuscript (Lines 218-230):

“Here, we compare the long-term trends in the $\delta^{18}\text{O}$ records with temporal changes in insolation and find that records loading strongly positive on PC1 follow summer insolation most closely, while those with a negative correlation to PC1 follow autumn insolation. The long-term trends of each $\delta^{18}\text{O}$ time series used for the PCA shows significant strong correlations with seasonal insolation for each site (Fig. 3G), further evidencing this link. We conclude that seasonal insolation is a main driver of $\delta^{18}\text{O}$ across Southeast Asia for both the SWM and NEM. $\delta^{18}\text{O}$ in all records becomes more negative when insolation is increased, however, autumn and summer insolation peak at different times during the Holocene, potentially causing the apparent anti-correlation (Fig. 3 and 4). Modern observations and model simulations indicate that decreased $\delta^{18}\text{O}$ in rainfall over Southeast Asia is linked to enhanced convection, more distal moisture source, and increased rainout along the trajectory (Wolf et al., 2020; Yang et al., 2016; Pausata et al., 2011). Thus, we propose that higher insolation leads to increased rainout and the moisture travel distance from the source, which results in decreasing $\delta^{18}\text{O}$ values in both the SWM and NEM. “

3. It's very interesting to see SWM and NEM show reverse response (reflected by the PC1 of $\delta^{18}\text{O}$) on orbital timescale, whereas coherent rainfall pattern across Southeast Asia was found on millennial timescales (reflected by the PC1 of $\delta^{13}\text{C}$). It's worth to further investigating the underlying mechanisms of the different regional circulation ($\delta^{18}\text{O}$) and local rainfall ($\delta^{13}\text{C}$) on different timescales. But I don't think the mid-Holocene time-slice model simulation results alone are sufficient to explain the difference in the response of $\delta^{18}\text{O}$ and rainfall on different timescales.

We generally agree with the reviewer, however, isotope-enabled transient model simulations are, to our knowledge, not yet available for the time period covered by our proxy record. We would also like to highlight that the mid-Holocene simulations were not aimed at understanding drivers for the entire Holocene, as stated in lines 285-291: *“Crucially, the dry conditions observed in the proxy records between ~5 to ~3 ka are evident in both PC1 and PC2, suggesting strong coherence between the sites during this time (Fig. S7). [...] In the next step we will investigate the forcing mechanisms of this dry period and their effects on the regional and local aspects of the Southeast Asian monsoon.”*

Instead, we found that the local rainfall proxy records shared most similarities during this drought period and aimed at investigating potential drivers for this period specifically. Our simulations indicated that a shift in the Pacific Walker Circulation due to vegetation changes over the Saharan region caused a decrease in rainfall over the entire region of SE Asia. Thus, at least for this time period, we argue that the decoupling of the local and regional monsoon lies in the different drivers, which includes the position of the Walker Circulation and seasonal insolation, respectively.

However, we agree that further research should focus on extending simulations, especially transient ones to investigate the decoupling of local and regional components of the SWM and NEM.

We adjusted the subtitles “Holocene monsoon coherence across Southeast Asia” and “Monsoon coherence during the mid-Holocene” for clarification.

4. It's not clear to me why the authors use “MHORB-MHGS” to discuss the mechanisms of $\delta^{18}\text{O}$ and precipitation changes during the mid-Holocene observed from proxy records.

The MHORB simulations incorporate mid-Holocene insolation, greenhouse gases, vegetation cover, and dust load, and MHGS simulations have a prescribed vegetated ‘Green Sahara’

(GS), additionally” (Line 287-290). It’s not clear to me what climate forcings would “MHORB-MHGS” indicate and why the authors use the difference between the ORB and GS runs to identify the mechanisms driving mid-Holocene anomalies. Another issue related to this is the argument about reduced Saharan vegetation could contribute to the mid-Holocene lower average rainfall observed in proxy records (Line 331-338, Fig. S9B). It’s not clear to me why “MHORB-MHGS” can specifically reveal the influence of Saharan vegetation. I would suggest the authors use the MHORB minus preindustrial (or early Holocene) to investigate the contribution of orbital forcing on mid-Holocene $\delta^{18}\text{O}$ and precipitation anomalies, and (MHGS - MHORB) minus preindustrial (or early Holocene) to evaluate the influence of Green Sahara.

The MHORB has the vegetation prescribed as pre-industrial as is in the standard PMIP protocol; however, such vegetation is unrealistic to represent the Green Sahara. On the other hand the MHGS include a more realistic representation of the Green Sahara. Hence, we use the MHORB-MHGS to isolate the effects of vegetation and dust forcing during the mid-Holocene and investigate the transition period leading to a dry Sahara but still with higher summer insolation than the pre-industrial. By plotting the difference between the MHORB run (standard protocol; pre-industrial Sahara) and MHGS (changes in vegetation and dust loading), we can isolate the effect of dust loading only on the spatial pattern in $\delta^{18}\text{O}$.

We then compare these results to our simulations for precipitation (also isolating the GS effect only) and find that a similar decoupling between simulated $\delta^{18}\text{O}$ and simulated rainfall persists. E.g., the annual mean precipitation for all 3 models (model mean Fig. 6 bottom), shows a decrease in precipitation across SE Asia, whereas iCESM shows a decrease in $\delta^{18}\text{O}$ only for north-eastern parts of SE Asia. Thus we argue that the decoupling of the local and regional components of the monsoon is consistent in model and proxy data, even when vegetation is simulated more realistically. Further, we identify the position of the Pacific Walker Circulation as a main driver of local rainfall coherence in SE Asia (Fig. S5), at least for the mid-Holocene. This finding allows us to conclude that $\delta^{18}\text{O}$ is mainly driven by insolation changes (as found in the previous section) and local rainfall by the position of the Walker Circulation during the mid-Holocene drought period.

To clarify our approach we added: “In summary, we compare runs simulating the vegetation over the Sahara as pre-industrial with simulations using a Green Sahara, to isolate the effect of dust loading (MHORB-MHGS).” Lines 314-316.

Other comments:

Fig. 1: (1) Please change “mean daily precipitation per month [mm/day]” in Fig. 1A and 1B to “mean monthly precipitation rate [mm/day]”. (2) Please change TH to TD (Line 108).

done

Fig. 2: Please use appropriate statistical methods to detect shifts in $\delta^{18}\text{O}$, $\delta^{13}\text{C}$, and Mg/Ca. Please make sure the cubic fit in Fig. 2E and 2F can reveal the long-term trend in $\delta^{18}\text{O}$ from TM and TD, respectively.

done

Line 162-166: Please show changes in autumn and summer insolation in Fig. 2 to back up this statement.

We have added another panel to figure 3 (below) showing a comparison of autumn and summer insolation to the cubic fits of the $\delta^{18}\text{O}$ time series.

Fig. 3: (1) Please add the time series of PC2 of $\delta^{18}\text{O}$ orig, $\delta^{18}\text{O}$ corr, and d^{13}C . At this time figure 3 has become quite busy and we decided to add the PC1s and PC2s, as well as a comparison plot in a new figure (Fig. 4; main text). Supplementary Fig. 6 has become obsolete and we removed it.

Please reverse the color bar for EOF, so that the warm color indicates positive loading, and the cold color indicates negative loading.

done

(2) In Fig. 3A-3F, please change "PC1" and "PC2" to "EOF1" and "EOF2".

done

Reviewer #3 (Remarks to the Author):

The authors present a new isotope and trace element record from speleothems from Vietnam. They claim that it is the first reconstruction of winter monsoon for the Holocene period, which is important for understanding the continent's paleoclimate. The data has high quality, and the comparisons with climate models were also addressed to reinforce their interpretation. The paper is well written, and this novel has the potential to be published in Nature Communications. However, I see a major problem with their correction on the $\delta^{18}\text{O}$ record. This correction compromises the quality of the data and the interpretation on the manuscript. See my concerns about this correction below. Apart from this, I have only minor comments to improve the manuscript.

1) The correction is based on the approach presented by Deininger et al. (2021), which shows that PCP can also affect the $\delta^{18}\text{O}$ in stalagmites. This is something that so far has been ignored in the literature. However, the implication of the PCP on $\delta^{18}\text{O}$ presented in Deininger et al. is mainly theoretical and still needs further validation before being applied to speleothem data records. Furthermore, no calibration studies with speleothem were done before.

To our knowledge, there is indeed no calibration study available, which investigated a direct link between $\delta^{18}\text{O}$ and Mg/Ca during carbonate precipitation. However, in individual studies it was shown, that $\delta^{18}\text{O}$ values are increasing with ongoing precipitation and there is a common agreement, that the Mg/Ca ratio is doing the same. See, e.g., the recent study by Stoll et al., in *Climate of the past*, which is still under review, but freely available), in which they also apply a first step to correct for the effect of PCP. We also oppose the comment of the 'mainly theoretical' nature of this effect. This process has been shown to be important in many laboratory experiments (e.g., Day and Henderson, 2012, Polag et al., 2010, Hansen et al., 2019, Mickler et al., 2004), where the stable isotope composition is always increasing along the path of carbonate precipitation.

Further we are attaching the recently accepted paper by 2 of our co-authors as additional supplemental material, which can also be found here: Skiba, V. and Fohlmeister, J., 2023. Contemporaneously growing speleothems and their value to decipher in-cave processes—A modelling approach. *Geochimica et Cosmochimica Acta*, 348, pp.381-396.

This study investigated the effect of PCP on speleothem $\delta^{18}\text{O}$ and $\delta^{13}\text{C}$ using an Rayleigh-based isotope evolution model which also accounts for O and C exchange processes. They successfully tested the model on a large dataset (SISAL v2) of speleothems. Our PCP

correction approach is based on the conclusions made in their study and in other earlier studies (e.g. Dreybrodt & Scholz 2011).

2) For this correction, the authors assume temperature and CO₂ in the soil to be constant during the Holocene period, which is unrealistic. Temperature changes were small but not constant. Slight elevations of temperature associated with drier periods can significantly increase evaporation with all associated effects on the isotopes. But the big problem is with regard the CO₂. The monitoring of soil CO₂ over two years already shows variability between 430 ppmv and 1580 ppmv. This variability was probably more significant during the Holocene. Furthermore, long periods of drier or wetter conditions can affect the vegetation and soil above the cave, changing the CO₂ concentration in the soil, which is the primary source of the cave's CO₂. The proxies presented in this paper present high variability, indicating significant rainfall variability above the cave, which will also affect the CO₂ concentration in the soil.

We are grateful for this suggestion and have conducted further simulations for our time series using varying parameter settings, including a range in cave air temperatures based on reconstructions from Borneo (Løland et al., 2022), which identified a range of $\pm 2^\circ\text{C}$ in the cave over the last 8 kyrs. We also add a range of CO₂ estimates to simulate potential changes in $p\text{CO}_2$ ranging from 260 to 3000 ppmv. In the table below we list the effect of these parameters on $\delta^{18}\text{O}$ during 6-4 ka. This period is likely to have seen the highest amounts in PCP over the last 8 kyrs. We find that changes in cave air temperature have only small effect on mean $\delta^{18}\text{O}$ ($\pm 0.05\text{‰}$), whereas cave air CO₂ variability affects the CO₂ up to $\sim -1\text{‰}$ (see table below). We consider our initial simulation with 426 ppmv and 21°C the 'main' simulation and the 12 other the 'uncertainties'.

Table 1: Mean $\delta^{18}\text{O}$ minus mean of main correction (460 CO₂ and 21° T) (between 6 and 4 ka BP).

$p\text{CO}_2$ (ppmv)	260	1000	2000	3000
T (°C)				
19	0.05	-0.15	-0.71	-1.11
21 (modern)	3.76e-05	-0.17	-0.69	-1.08
23	-0.05	-0.19	-0.67	-1.05

We have also added more text (below) and figure (Figs. 3G and 4) discussing the effect of our PCP correction on our results to the main text (lines 231-242):

“Interestingly, the majority of our PCP-corrected time series (presented as the cubic fits, Fig. 3G) closely follows insolation, however, those corrections with high cave air $p\text{CO}_2$ (3000 and 2000 ppmv) are deviating the most. Values much higher than 1600 ppmv (highest modern cave air $p\text{CO}_2$ is ~ 1600 ppmv) are unlikely to have been reached over the Holocene, due to the cave being well-ventilated and changes in atmospheric CO₂ over the Holocene only varied between ~ 240 to 270 ppmv (by 30 ppmv) (Petit et al., 1999). While unlikely, it should be noted that an increase in cave air $p\text{CO}_2$ to 2000 and 3000 ppmv alters the long-term trend of $\delta^{18}\text{O}$ and reduces the correlation between the cubic fit of $\delta^{18}\text{O}$ and SON insolation to $R = -0.69$ (for 3000 ppmv, 21°C), compared to $R = -0.92$ (for 426 ppmv, 21°C). Considering the uncertainties in our novel correction approach and recognizing that cave environment processes are simplified, we find that our results are a promising step in accounting for PCP influence on speleothem $\delta^{18}\text{O}$, which can enable the separate investigation of local and regional dynamics in our proxy record.”

3) The authors use $\delta^{13}\text{C}$ and Mg as a “pure” proxy for PCP, which is unrealistic. I agree that both are proxies for PCP, but PCP is not the only force on these proxies. In using Mg to correct $\delta^{18}\text{O}$, the authors are not only removing the PCP signal but also adding noise to the $\delta^{18}\text{O}$ signal.

We agree with the reviewer, but would like to highlight that we only focus on the Holocene trend and do not use / interpret the short-term variability of the record, due to the limitations of our PCP correction. Therefore we focus on the local proxies ($\delta^{13}\text{C}$ and Mg/Ca) and the PCA simulations for our mid-Holocene interpretation. Our statistics (PCA and trend analysis) aim to filter any noise our uncertainties related to the local proxies. Further, we base our conclusions on a suite of 5 speleothem records from SE Asia to provide a more robust interpretation for our findings.

4) In this correction, the author assumes that all correlation between $\delta^{18}\text{O}$ and Mg and $\delta^{13}\text{C}$ is due to PCP. However, other factors provide a similar correlation between these proxies other than PCP. Therefore, in removing this dependency from the $\delta^{18}\text{O}$, the author also removes additional information. For example, during a strong monsoon, the “amount effect” provides more negative values to the $\delta^{18}\text{O}$, but the high rainfall during this event also decreases the PCP leading to lower values of Mg and $\delta^{13}\text{C}$. Therefore amount effect and rainfall variability also drive $\delta^{18}\text{O}$, $\delta^{13}\text{C}$, and Mg in the same direction. Therefore, using the Mg/Ca and correcting the $\delta^{18}\text{O}$, the authors are not just removing the PCP effect on the $\delta^{18}\text{O}$ but also the amount effect.

There is strong evidence that the amount effect is not controlling $\delta^{18}\text{O}$ at our cave sites (Wolf et al., 2020, Yang et al., 2016, Pausata et al., 2011). Data from the local GNIP station shows that there is no correlation between monthly $\delta^{18}\text{O}$ and rainfall amount (for the last 5 years; Fig. 1 main manuscript; Wolf et al., 2020). Instead, it has been shown that the reversal of winds, changing the location of moisture source, drives the seasonal and interannual variability of $\delta^{18}\text{O}$ in central Vietnam. Thus, a correlation between $\delta^{18}\text{O}$ and $\delta^{13}\text{C}$ /Mg based on the amount effect is not to be expected in our record.

Additionally our correction does not assume that the entire $\delta^{18}\text{O}$ record is PCP affected and has implemented to use the lowest Mg/Ca values as equal to zero PCP. Thus, the initial $\delta^{18}\text{O}$ value of the water, including any variability from rainfall distribution, should be reflected better after the PCP correction.

5) The authors argue that they are removing the effect of PCP on the $\delta^{18}\text{O}$. But the corrected record has much higher variability than the original record, making me believe that this “correction” adds more variability and noise to the records than filter PCP.

Again we would like to highlight that we are not using variability or the amplitude of individual peaks in our interpretation, but rather focus on the long-term trends, which is extracted by the cubic fits and PCA. Therefore, despite the limitations of our corrections, such as estimated cave parameters, we have confidence in our conclusions on the co-evolution of $\delta^{18}\text{O}$ in SWM and NEM rainfall in Southeast Asia.

Minor comments:

Fig.1: How the amount-weighted $\delta^{18}\text{O}$ was calculated? Were monthly averages used?

yes monthly averages were used (no daily data is available) to calculate the amount-weighted $\delta^{18}\text{O}$ with: $\delta_{\text{each month}} * (\text{precip_month} / \text{precip_annual_total})$

Fig2: I suggest including the name of the caves in the figure or at least on the captions. It isn't enjoyable to go to another figure to see the name of the records presented in this figure.
done

Fig. 2: it is difficult to distinguish the scales at the y-axes between D and E.
adjusted

The authors also measured the Sr/Ca and Ba/Ca with laser ablation. Why were these data not shown in the paper?

The data is shown in figure S1 as part of the PCA plot (S1F). We have also added the time series in Fig. S1C for clarification.

Line 258: The authors say "This suggests that the first PC derived from stalagmite $\delta^{13}\text{C}$ records and a lake record from Southeast Asia likely share a common forcing throughout the Holocene...".

According to the authors, $\delta^{13}\text{C}$ in the stalagmites is connected to the local hydrology due to the PCP process. However, PCP is absent in lakes. $\delta^{13}\text{C}$ in lakes is related to the isotopic composition of the organic matter/vegetation. If $\delta^{13}\text{C}$ from stalagmite and lakes correlates, this is suggestive that the $\delta^{13}\text{C}$ from speleothems is also affected by the $\delta^{13}\text{C}$ from the vegetation (something ignored in this manuscript) and not that the $\delta^{13}\text{C}$ from the lakes is affected by PCP, which is impossible. How do the authors explain it?

We apologise for this confusion. The lake record is using the Ti content as a proxy for winter monsoon strength not $\delta^{13}\text{C}$, noted in lines (235-236 and 256-257). However, we recognise that the figure title was misleading and have renamed it to $\delta^{13}\text{C} + \text{Ti}$.

REVIEWER COMMENTS

Reviewer #2 (Remarks to the Author):

Dr. Woft et al. have addressed most of my main concerns that were detailed in my earlier review. In particular, I think the incorporation of trend/change point detection and the comparison between the cubic fits for $\delta^{18}O$ records and seasonal solar insolation strengthen the manuscript. However, there are still some aspects that are unclear from my perspective.

The simulated influence of the Green Sahara on rainfall and $\delta^{18}O$, and its comparison with proxy records, is still not clear to me. The authors have clarified that MHORB has prescribed pre-industrial vegetation, while MHGS uses a more realistic representation of the Green Sahara. Therefore, I assume that "MHORB - MHGS" represents the influence of "reduced Saharan vegetation and increased dust." If that is the case, Fig. 6 (PRECT: MHORB - MHGS) demonstrates that the reduced Saharan vegetation and increased dust during MH would lead to drier conditions in Southeast Asia, as described in Lines 355-362. In other words, the mid-Holocene Green Sahara (a vegetated and less dusty Sahara) would lead to wetter conditions in Southeast Asia, which seems to contradict the widespread mid-Holocene drought in paleoclimate proxy records (EOF1 and PC1 of $\delta^{13}C + Ti$). Addressing this point is important for establishing the linkage between the Green Sahara and monsoon rainfall in Southeast Asia.

Related to the above point, please further explain how the spatial pattern of MHORB-MHGS $\delta^{18}O$ shown in Fig. 5B supports the different Holocene trends in SWM and NEM $\delta^{18}O$ (Lines 324-331). The MHORB-MHGS $\delta^{18}O$ values increase only in northern MSEA and Sumatra, corresponding to reduced Sahara vegetation and increased dust (MHORB-MHGS). But as shown in Fig. 3G, all proxy records across Southeast Asia show lower $\delta^{18}O$ during the mid-Holocene relative to preindustrial.

The discussion on the linkage between insolation and changes in SWM and NEM has been improved in the current version. I would suggest the authors further test the mechanism of the insolation influence on SWM and NEM by comparing MHORB $\delta^{18}O$ and preindustrial $\delta^{18}O$ in iCESM (isolating the influence of solar forcing).

Other comments:

Line 227-230: More evidence is needed here to support the claim that changes in solar insolation during the Holocene can influence $\delta^{18}O_p$ in the SWM and NEM by altering moisture source and rainout along the trajectory.

Line 242: Please delete the quotation mark at the end of this sentence.

Line 249: Please add "(LH)" after "Lake Huguang Maar."

Line 254: I would suggest replacing "Information contained in the remaining trends (PC2)" with "The

overall positive loading on the PC2 of d18O" to make this sentence clearer.

Line 257-258: Please change "Fig. 3E" to "Fig. C4" or change "PC2" to "EOF2".

Line 316: Please add "during the Mid-Holocene" to the end of this sentence to avoid potential confusion.

Line 372: I would suggest changing "accurate" to "reliable."

Fig. 4:

(1) Please correct the title of Fig. 4C to "PC2 of d18O and PC1 of d13C."

(2) Please explain the meaning of the dashed lines in Fig. 4A and 4B in the figure caption.

(3) Please add "of d18Op" after "The uncorrected PC2."

Reviewer #3 (Remarks to the Author):

I'm still skeptical about the correction on the d18O values. But the authors presented good arguments to justify these corrections in the manuscript. Furthermore, all my other minor suggestions were incorporated. Therefore, I recommend the publication in Nature Communication without further changes.

Best regards,

Valdir F. Novello

Response to referees

Reviewer #2 (Remarks to the Author):

Dr. Wolf et al. have addressed most of my main concerns that were detailed in my earlier review. In particular, I think the incorporation of trend/change point detection and the comparison between the cubic fits for $\delta^{18}\text{O}$ records and seasonal solar insolation strengthen the manuscript. However, there are still some aspects that are unclear from my perspective.

The simulated influence of the Green Sahara on rainfall and $\delta^{18}\text{O}$, and its comparison with proxy records, is still not clear to me. The authors have clarified that MHORB has prescribed pre-industrial vegetation, while MHGS uses a more realistic representation of the Green Sahara. Therefore, I assume that "MHORB - MHGS" represents the influence of "reduced Saharan vegetation and increased dust." If that is the case, Fig. 6 (PRECT: MHORB - MHGS) demonstrates that the reduced Saharan vegetation and increased dust during MH would lead to drier conditions in Southeast Asia, as described in Lines 355-362. In other words, the mid-Holocene Green Sahara (a vegetated and less dusty Sahara) would lead to wetter conditions in Southeast Asia, which seems to contradict the widespread mid-Holocene drought in paleoclimate proxy records (EOF1 and PC1 of $\delta^{13}\text{C} + \text{Ti}$). Addressing this point is important for establishing the linkage between the Green Sahara and monsoon rainfall in Southeast Asia.

Thank you for this comment! Yes, it is correct that $\text{MH}_{\text{ORB}} - \text{MH}_{\text{GS}}$ represents the reduced Saharan vegetation and increased dust at the end of the MH, which lead to drier conditions in Southeast Asia. Thus, our simulations (Fig. 6) show that at the end of the Green Sahara increased dust load led to drier conditions across MSEA. This is well represented via the widespread drought (ca. 5-3.5 ka BP) in our proxy records in Figs. 4A and 4B and S2. Both the mean age of the onset of the GS termination (Fig. 1 below) and the start of the pronounced mid-Holocene droughts observed in our proxy data start around 5 ka. Here, we compare the conditions after the GS around 5 ka in the simulations to the drought also starting at 5 ka in our proxy records. We apologize for the confusion and have updated the main text as: "Here, we compared the changes in the local proxy record before (8-5 ka BP) and after (5-3.5 ka BP) the end of the Green Sahara, where all proxy records used in the MC-PCA suggest drier conditions during 5 to 3.5 ka BP." Lines 347-349.

The figure below from Griffiths et al., 2020 (Ref 32 in main text) shows the timing of the termination of the GS.

Figure 1: Caption from Griffith et al., 2020: “a Northern Laos composite $\delta^{13}\text{C}$ record (black line) and 1σ uncertainty (gray shading) from Tham Doun Mai speleothems. b–d $\delta\text{D}_{\text{max}}$ records from marine core P178–15P (Gulf of Aden)³, Lake Challa⁴⁰, and Lake Tanganyika⁴¹. e Percent dolomite (light purple line) and carbonate (pink line) (expressed as standardized Z-scores) from the Gulf of Oman³³. Also shown is the dust record (aquamarine line) from Mt. Kilimanjaro ice core KNIF338. f Ca/Ti record of dust deposition in the Nile Delta⁴². g Color-coded (cyan: Lake Challa; orange: Lake Tanganyika; black: Tham Doun Mai; blue: Gulf of Aden) probability density function (PDF) output from the Bayesian change-point algorithm³⁹. Vertical color bar indicates the transition from a wet to a dry Sahara between 5.5 and 3.5 ka^{2,3}.”

Related to the above point, please further explain how the spatial pattern of MHORB-MHGS $\delta^{18}\text{O}$ shown in Fig. 5B supports the different Holocene trends in SWM and NEM $\delta^{18}\text{O}$ (Lines 324–331). The MHORB-MHGS $\delta^{18}\text{O}$ values increase only in northern MSEA and Sumatra, corresponding to reduced Sahara vegetation and increased dust (MHORB-MHGS). But as shown in Fig. 3G, all proxy records across Southeast Asia show lower $\delta^{18}\text{O}$ during the mid-Holocene relative to preindustrial.

There was an incorrect label for the figure reference in this section. We apologize and have changed Fig. 3G to Fig. 3A and C in Line 324. Here, we aimed at comparing the spatial pattern: negative loading on Vietnam and Borneo vs positive loading on China, Laos and Sumatra, compared to the model simulation $\delta^{18}\text{O}$ SON, which has a similar spatial pattern. Plotting MHORB-MHGS removes the effects of insolation, thus we cannot compare the temporal trends in the proxies to the simulation and instead compare the spatial pattern of the simulations to the spatial pattern in the timeseries (Fig. 3A and C in the main text). Since the end of the GS causes an increase in $\delta^{18}\text{O}$ over northern MSEA and Sumatra only we conclude that the pattern of an inverse relation between SWM and NEM is enhanced by the dust loading via an additional increase in $\delta^{18}\text{O}$ for SWM records, while NEM is at its lowest values of the Holocene.

The discussion on the linkage between insolation and changes in SWM and NEM has been improved in the current version. I would suggest the authors further test the mechanism of the insolation influence on SWM and NEM by comparing MHORB d18O and preindustrial d18O in iCESM (isolating the influence of solar forcing).

Thank you for this comment. Please see the figure below, which was added to the supplement as Fig. S11. We can see that by considering the changes in vegetation and dust the differences in isotopic composition between the mid-Holocene and pre-industrial are enhanced and in better agreement with proxy records than when considering orbital forcing only. Unfortunately, not all proxy records extent towards modern times and thus we cannot directly compare the simulated changes with the paleoclimate archives.

Figure 2: iCESM simulation of the change in $\delta^{18}\text{O}$ of annual precipitation for mid-Holocene orbital forcing – preindustrial (PI; 1850 CE) conditions (left) and mid-Holocene orbital + the Green Sahara forcing – PI (right). Stippling highlight changes significantly different at the 5% level using a t test.

Other comments:

Line 227-230: More evidence is needed here to support the claim that changes in solar insolation during the Holocene can influence d18Op in the SWM and NEM by altering moisture source and rainout along the trajectory.

Our simulations do show a change in $\delta^{18}\text{O}$ p in the SWM and NEM (Fig. 5, main text); however, our simulations do not include moisture tracking, so we cannot verify it through our experiments. Nevertheless, the influence of the insolation on speleothem $\delta^{18}\text{O}$ has been shown in many speleothem records from East Asia and other monsoon regions, as well as that the effects of moisture source and rainout dynamics control modern $\delta^{18}\text{O}$ p in MSEA, please see references 22, 23, 58 in main text and other studies (e.g. Zhang et al., 2018; Hu et al., 2019 (using iCESM and speleothems); He et al., 2021). The role of moisture source dynamics affecting $\delta^{18}\text{O}$ p in iCESM has also been identified for the mid-Holocene in Tabor et al. 2020 (62) and Thompson et al. 2021 (83). Thus, we propose that moisture source dynamics and rainout changed based on insolation, which is reflected in our proxy record. Such changes are further enhanced by vegetation and dust changes

which further alter the intensity of the Southeast Asian monsoon system. We agree that additional studies are required to back up this statement, however, this is beyond the scope of our paper.

Line 242: Please delete the quotation mark at the end of this sentence.

done

Line 249: Please add "(LH)" after "Lake Huguang Maar."

done

Line 254: I would suggest replacing "Information contained in the remaining trends (PC2)" with "The overall positive loading on the PC2 of d18O" to make this sentence clearer.

done

Line 257-258: Please change "Fig. 3E" to "Fig. C4" or change "PC2" to "EOF2".

done

Line 316: Please add "during the Mid-Holocene" to the end of this sentence to avoid potential confusion.

done

Line 372: I would suggest changing "accurate" to "reliable."

done

Fig. 4:

(1) Please correct the title of Fig. 4C to "PC2 of d18O and PC1 of d13C."

done

(2) Please explain the meaning of the dashed lines in Fig. 4A and 4B in the figure caption.

done

(3) Please add "of d18Op" after "The uncorrected PC2."

done

Reviewer #3 (Remarks to the Author):

I'm still skeptical about the correction on the d18O values. But the authors presented good arguments to justify these corrections in the manuscript. Furthermore, all my other minor suggestions were incorporated. Therefore, I recommend the publication in Nature Communication without further changes.

We are grateful for the provided feedback and ideas!

References

Hu, J., Emile-Geay, J., Tabor, C., Nusbaumer, J., & Partin, J. (2019). Deciphering oxygen isotope records from Chinese speleothems with an isotope-enabled climate model. *Paleoceanography and Paleoclimatology*, 34(12), 2098-2112.

He, C., Liu, Z., Otto-Bliesner, B. L., Brady, E. C., Zhu, C., Tomas, R., ... & Bao, Y. (2021). Hydroclimate footprint of pan-Asian monsoon water isotope during the last deglaciation. *Science Advances*, 7(4), eabe2611.

Zhang, H., Griffiths, M.L., Chiang, J.C., Kong, W., Wu, S., Atwood, A., Huang, J., Cheng, H., Ning, Y. and Xie, S., 2018. East Asian hydroclimate modulated by the position of the westerlies during Termination I. *Science*, 362(6414), pp.580-583.

REVIEWERS' COMMENTS

Reviewer #1 (Remarks to the Author):

The authors have well addressed all of my comments, and I appreciate their efforts. I fully support the publication of this paper in Nature Communications.